 SciPost Phys. Lect. Notes 55 (2022)

# Dark Matter direct detection of classical WIMPs

**Jodi Cooley**⋆

Department of Physics, Southern Methodist University, Dallas, Texas 75275, USA

⋆ cooley@physics.smu.edu

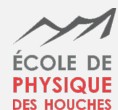

*Part of the Dark Matter*
*Session 118 of the Les Houches School, July 2021*
*published in the Les Houches Lecture Notes Series*

## Abstract

One of the highest priorities in particle physics today is the identification of the constituents of dark matter. This manuscript is a supplement to pedagogical lectures given at the 2021 Les Houches Summer School on Dark Matter. The lectures cover topics related to the direct detection of Weakly Interaction Massive Particle (WIMP) dark matter, including the distribution of dark matter, nuclear scattering, backgrounds, planning and designing of experiments, and a sampling of planned and ongoing experiments.

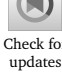

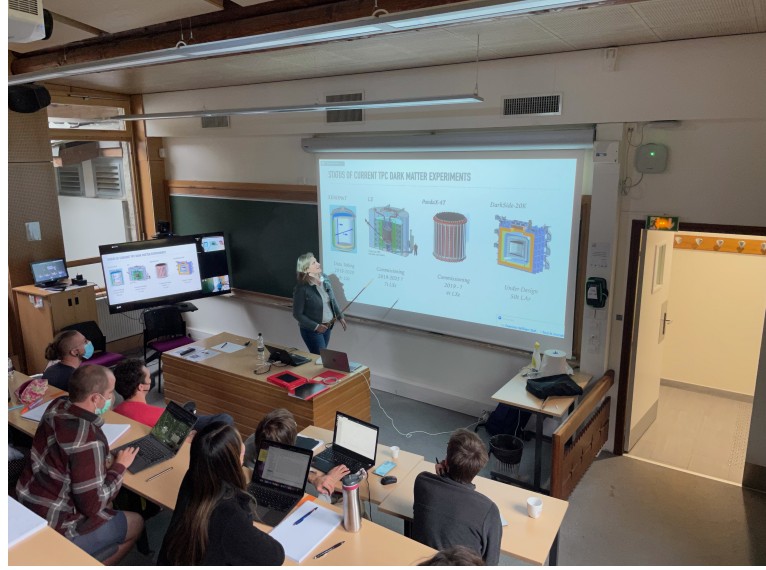

# Contents

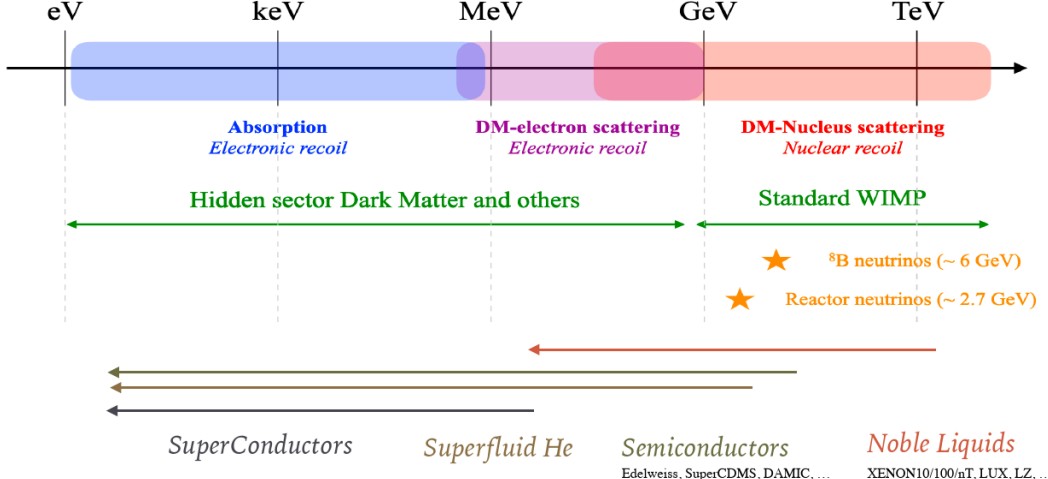

Figure 1: A logarithmic plot of possible masses for dark matter particles, as well as general techniques and specific experiments in various ranges that have, are, or will attempt to constrain dark matter in that region. In these lectures, the focus is on WIMPs whose masses are typically greater than $\sim$1 GeV/c$^2$.

# 1 Introduction

Detecting dark matter by *direct means* - that is, observing the effect of dark matter as a particle in a controlled experiment that isolates its interactions - has proven to be a great challenge. Given what we know about it from astrophysics and cosmology - that it interacts with certainty only through gravitational influence - poses perhaps the greatest challenge to experimentalists. We are not assured of its interactions by any other known force and should it interact by forces *unknown,* we do not know what character those forces might possess. As such, there is a vast phase space of theoretical consideration that lies open to experiment.

To design a dedicated direct dark matter detection experiment, one needs to factor in a number of considerations:

- What may be the rate of dark matter passing through the experimental apparatus?

- What may be the rate of interaction between dark matter and baryonic matter?

- What known interactions and particles might confound the observations?

- What will be the potential totality of experimental signatures of dark matter interacting with the instrumentation?

It's possible, of course, that we've already detected dark matter. Experiments capable of observing the scattering of a weakly interacting, electrically neutral particle have existed for decades. The scattering process might happen off an atomic nucleus or electrons, or both. In principle, however, the challenge to the experimentalist lies in the lack of constraints on just what kind of particle we are hunting, perhaps most well-represented by the sheer lack of constraint on the mass of such a particle. It could be so low in mass that its effective de Broglie wavelength spans crystal lattices, labs spaces, or whole planets. On the other hand, it could be so heavy that it ejects heavy nuclei wholesale from solids, liquids, or gases. A good example of the challenge, and categories of experiments that have, are, or will rise to meet that challenge, is shown in Fig. 1.

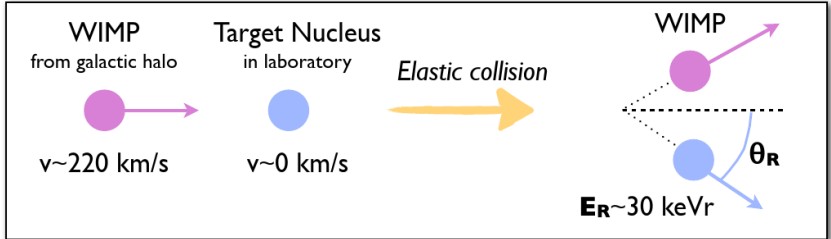

Figure 2: A schematic of dark matter (WIMP) elastic scattering off a target nucleus.

## 2 Dark Matter Distributions

The search for dark matter's particle constituents is for naught if the distribution of dark matter in our local region of the Milky Way is non-existent. I will first focus on the distribution of relic Weakly Interacting Massive Particle (WIMP) dark matter in the Milky Way halo. Understanding the distribution is a crucial ingredient in considerations for the construction of experiments and interpretation of their (non-)observations.

### 2.1 A Simplified Model of WIMP Distribution in the Milky Way

WIMPs are assumed to be organized in isothermal spherical halos with a Gaussian velocity distribution, owing to random fluctuations about some mean velocity. This is all encapsulated in a Maxwell-Boltzmann model of the halo, with a "Maxwellian" velocity distribution,

$$f(\vec{v}) = \frac{1}{\sqrt{2\pi}\sigma} e^{-|\vec{v}|^2/(2\sigma^2)}, \tag{1}$$

where $\sigma$ is the WIMP velocity dispersion. The local circular speed, $v_c$, is defined as the motion of our solar system about the center of the galaxy. The speed dispersion, $\sigma$ is related to the local circular speed by

$$\sigma = \sqrt{\frac{3}{2}} v_c, \tag{2}$$

where the local circular velocity is the speed of the our solar system about the center of the galaxy, taken as $v_c = 220$ km/s.

The density profile of the sphere is $\rho(r) \propto r^{-2}$ and $\rho_0 = 0.3$ GeV/c$^2 \cdot$ cm$^3$, where the local density of dark matter near Earth is taken to be the average of a small volume within a few hundred parsecs of the Sun. There are two approaches to measuring the local dark matter density [1]. The first involves the use of the vertical kinematics of stars near the Sun [2–5] and the second involves extrapolation from galactic rotation curves [6]. In our region of the galaxy, particles with speeds greater than the escape velocity, $v_{esc}$ are not gravitationally bound. Hence, the speed distribution needs to be truncated and will cut off at $v_{esc} = 650$ km/s.

It is interesting to then think about what will be the population of WIMPs in your work area. We have the local dark matter density in our region of the Milky Way. You need to next select your "favorite" mass for a WIMP candidate. I will choose one lighter species, $m_{\chi_1} = 5$ GeV/c$^2$ and one heavier species, $m_{\chi_2} = 100$ GeV/c$^2$. What then will be their respective number densities? These are

$$n_1 = \rho_0/m_{\chi_1} = 6 \times 10^{-2} \text{ cm}^{-3}, \tag{3}$$

$$n_2 = \rho_0/m_{\chi_2} = 3 \times 10^{-3} \text{ cm}^{-3}. \tag{4}$$

Using these number densities, we can then take a room with a certain volume and determine at any moment in time what are the number of particles in the room. Let's choose a more familiar volume - that of a large (2 liter) water or soda bottle. In that case,

$$N_1 = n_1 V_{bottle} = 120, \tag{5}$$
$$N_2 = n_2 V_{bottle} = 10. \tag{6}$$

Detector volumes will range from compact (liters in volume) to large-scale (room-sized or larger in volume). It is clear that, given what we know about the local dark matter halo, there is a good chance for dark matter to be passing through such volumes. What will control the rate of actual detection are two factors: the fundamental strength of the interaction between baryonic and dark matter (which may be weak-scale, but since that was determined from a very simplistic hypothesis, perhaps not) and the acceptance and efficiency of an active detection medium for the energy deposited by these interactions.

## 3 Nuclear Scattering of Dark Matter

Designing an experiment to directly detect dark matter means establishing immediately a fundamental hypothesis: *dark matter is not only gravitationally interacting*. A popular assumption is to consider some kind of weak-interaction-like framework (even including the actual standard model weak interaction, involving an exchange of weak bosons or the Higgs boson). This is couched in the language of "WIMP searches," referring to the weakly interacting massive particles inspired by freeze-out considerations discussed elsewhere [7]. The term "WIMPs" refer to one candidate class of dark matter particles that can be heavy and weakly interacting.

The specific nature of the interaction, broadly speaking, doesn't matter here. All that matters is that it is stronger than the gravitational interaction alone which would be so feeble in a subatomic terrestrial experiment as to be totally undetectable.

Our consideration of galaxy dynamics and ideal gas models has led us to conclude that a typical dark matter particle velocity in the Milky Way's dark matter halo at our radius from the galactic center would be about 220 km/s, or $\beta = v/c = 7.34 \times 10^{-4}$. We can begin there - that a single dark matter particle has an average speed at this level. This certainly qualifies as non-relativistic, so we can treat a hypothetical scattering process using non-relativistic quantum mechanics and kinematics. Let us hypothesize a dark matter particle mass of 100 GeV/$c^2$ (about 100 times the proton mass). What kind of kinetic energy, on the scale of nuclear binding energy, are we expecting here?

$$K = \frac{1}{2}mv^2 = \frac{1}{2}m\beta^2 c^2 = 27 \text{ keV}. \tag{7}$$

The typical binding energy per nucleon in heavier nuclei is $6-8$ MeV ... this is no where near enough to fission nuclei. Therefore, we anticipate that interactions with the nucleus will be elastic in nature. Even for dark matter with mass of $\mathcal{O}(1TeV)$ we still expect elastic interactions. Although for this discussion, we will focus on elastic scattering, it should be noted that inelastic scattering is also possible [8].

The consequences of this kind of nuclear interaction are depicted in Fig. 2. This scattering process will impart a small amount of energy to the nucleus. While elastic, spin interactions may nevertheless play a role in the degree and kind of scatter and will need to be factored into scattering interpretations. In addition, it will be important to distinguish between *background scatters* - interactions in the nuclei caused by standard model interactions that have nothing to do with dark matter scattering.

## 3.1 WIMP-nucleus Scattering: Laboratory Frame

Let us approach the scattering process, using Fig. 2, in the *laboratory frame*: the frame in which the medium containing the nuclei is at rest. Given that the typical nuclear mass is going to be $m_N \sim \mathcal{O}(10 \text{ GeV}/c^2)$, and that the typical temperature of experimental material is going to be well below room temperature (to remove thermal noise from any measurements) at the level of $T \sim \mathcal{O}(1 \text{ K})$, the velocity of nuclei executing thermal motion is going to be

$$K = \frac{1}{2} m_N v_N^2 \approx k_B T \longrightarrow v_N \approx \sqrt{\frac{2 k_B T}{m_N}} \approx 10^{-7} \text{ m/s} \tag{8}$$

or a $\beta_N = v_N/c \approx 10^{-16}$. Since the dark matter is expected to be traveling at speeds consistent with the solar system's speed in orbit around the center of the galaxy ($220 \text{ km/s} \longrightarrow \beta_\chi \approx 10^{-3}$), we can safely approximate the nuclei to be at rest in the laboratory frame.

We can now develop the mathematics for this scattering. Given the low speeds involved, I will treat this as a non-relativistic collision subject to classical mechanics. In an elastic collision, both momentum (always conserved in closed and isolated systems) and kinetic energy (only conserved in this case) will be conserved. We can denote initial-state quantities without primes and final-state quantities with primes. Let $\vec{p}$ ($\vec{k}$) denote the dark matter (nucleus) momentum. Then:

$$\vec{p} + \vec{k} = \vec{p}' + \vec{k}' \tag{9}$$

$$\frac{p^2}{2m_\chi} + \frac{k^2}{2m_N} = \frac{(p')^2}{2m_\chi} + \frac{(k')^2}{2m_N}. \tag{10}$$

For the nucleus, $k = 0$ (the at-rest approximation). Thus:

$$\vec{p} = \vec{p}' + \vec{k}' \tag{11}$$

$$\frac{p^2}{2m_\chi} = \frac{(p')^2}{2m_\chi} + \frac{(k')^2}{2m_N}. \tag{12}$$

Let us further simplify this by placing the original dark matter particle entirely along the horizontal axis in Fig. 2. The nucleus will then be scattered relative to that axis at the indicated angle, $\theta_R$, so that $k'_\parallel = k' \cos \theta_R$ and $k'_\perp = k' \sin \theta_R$, where the parallel ($\parallel$) and perpendicular ($\perp$) symbols indicate direction relative to horizontal. Since there was no net momentum in the transverse direction in the initial state, and since momentum is conserved in each direction separately, it must be true in the final state that

$$\vec{p'}_\perp + \vec{k'}_\perp = 0 \longrightarrow \vec{p'}_\perp = -\vec{k'}_\perp, \tag{13}$$

which of course tells us that

$$p' \sin \phi = k' \sin \theta_R \longrightarrow \sin \theta_R = \frac{p'}{k'} \sin \phi \tag{14}$$

and relates the scattering angles to one another through the ratio of the final-state momentum magnitudes.

It is useful to define the *momentum transfer*, $\vec{q} = \vec{p}' - \vec{p}$. We can then identify that

$$q^2 = p^2 - 2\vec{p'} \cdot \vec{p} + (p')^2 \tag{15}$$

and that

$$\vec{q} = -\vec{k}'. \tag{16}$$

This let's use write the kinetic energy of the recoiling nuclear (also known as the *recoil energy*) as

$$E_R \equiv \frac{(k')^2}{2m_N} = \frac{q^2}{2m_N}. \tag{17}$$

Let us try to determine a relationship between the initial dark matter speed and the final recoiling nucleus speed. To do this, we will need to eliminate the final dark matter particle speed ($v'$) from our equations. From the conservation of kinetic energy,

$$\frac{1}{2}m_\chi v^2 = \frac{1}{2}m_\chi (v')^2 + \frac{1}{2}m_N (u')^2 \tag{18}$$

$$m_\chi v^2 = m_\chi (v')^2 + m_N (u')^2 \tag{19}$$

$$m_\chi^2 v^2 = m_\chi^2 (v')^2 + m_\chi m_N (u')^2 \tag{20}$$

$$m_\chi^2 (v')^2 = m_\chi^2 v^2 - m_\chi m_N (u')^2, \tag{21}$$

where $u$ is the speed of the recoiling nucleus. We can then employ the conservation of momentum,

$$m_\chi \vec{v} = m_\chi \vec{v'} + m_N \vec{u'} \tag{22}$$

$$m_\chi \vec{v'} = m_\chi \vec{v} - m_N \vec{u'} \tag{23}$$

$$m_\chi^2 (v')^2 = m_\chi^2 v^2 - 2m_\chi m_N \vec{v} \cdot \vec{u'} + m_N^2 (u')^2. \tag{24}$$

Now relate these two initial dark matter speed terms between the conservation of kinetic energy and momentum,

$$m_\chi^2 v^2 - m_\chi m_N (u')^2 = m_\chi^2 v^2 - 2m_\chi m_N \vec{v} \cdot \vec{u'} + m_N^2 (u')^2 \tag{25}$$

$$-m_\chi m_N (u')^2 = -2m_\chi m_N \vec{v} \cdot \vec{u'} + m_N^2 (u')^2 \tag{26}$$

$$2m_\chi m_N \vec{v} \cdot \vec{u'} = m_\chi m_N (u')^2 + m_N^2 (u')^2. \tag{27}$$

Let us now consider the case of *maximum energy transfer*, which will happen when the initial dark matter velocity is entirely collinear with the final nucleus velocity so that $\theta_R = 0$. In this case, $\vec{v} \cdot \vec{u'} = vu'$ and

$$2m_\chi m_N \vec{v} \cdot \vec{u'} = m_\chi m_N (u')^2 + m_N^2 (u')^2 \tag{28}$$

$$2m_\chi m_N v u' = m_\chi m_N (u')^2 + m_N^2 (u')^2 \tag{29}$$

$$2m_\chi m_N v u' = (m_\chi m_N + m_N^2)(u')^2 \tag{30}$$

$$2m_\chi m_N v = (m_\chi m_N + m_N^2) u' \tag{31}$$

$$\frac{v}{u'} = \frac{m_\chi m_N + m_N^2}{2m_\chi m_N}. \tag{32}$$

This can be transformed into a statement about the ratios of the kinetic energies of the initial dark matter particle and the final nucleus. This is a more relevant physical quantity since most detectors we will consider are *calorimeters*, meaning their primary sensitivity is to the amount

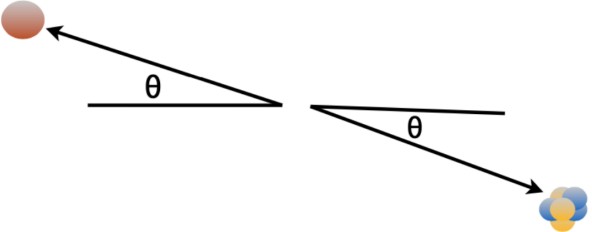

Figure 3: WIMP-nucleus scattering in the center-of-mass frame.

and kind of energy deposited in medium:

$$\frac{v}{u'} = \frac{m_\chi m_N + m_N}{2m_\chi} \tag{33}$$

$$\left(\frac{v}{u'}\right)^2 = \frac{(m_\chi + m_N)^2}{4m_\chi^2} \tag{34}$$

$$\frac{\frac{1}{2}m_\chi v^2}{\frac{1}{2}m_N(u')^2} = \frac{m_\chi(m_\chi + m_N)}{4m_{\chi^2}m_N} \tag{35}$$

$$\frac{\frac{1}{2}m_\chi v^2}{\frac{1}{2}m_N(u')^2} = \frac{(m_\chi + m_N)^2}{4m_\chi m_N}. \tag{36}$$

Since $E_R \equiv (k')^2/2m_N = \frac{1}{2}m_N(u')^2$,

$$E_R^{max} = \frac{1}{2}m_\chi v^2 \frac{4m_\chi m_N}{(m_\chi + m_N)^2}. \tag{37}$$

The quantity $\frac{m_\chi m_N}{(m_\chi + m_N)}$ is also known as the *reduced mass* of the system. If we put ourselves in the center-of-mass frame of the colliding system (a special point that, by conservation of momentum, is fixed in space), the reduced mass appears quite naturally. It can be though of as if a system of that singular mass $\mu$ was experiencing the collision, taking into account the relative motion of the original two bodies before the collision. Thus we can write:

$$E_R^{max} = 2m_\chi v^2 \frac{\mu^2}{m_\chi m_N} = 2\frac{\mu^2 v^2}{m_N}. \tag{38}$$

## 3.2 WIMP-nucleus Scattering: Center-of-Mass Frame

Let's explore the kinematics of this kind of scattering in more detail, but moving to the center-of-mass frame. While much of the notation in this section will look similar to the previous section, it is more of convenience than invariance. For example, the initial speed of the dark matter in the lab frame is given by the Earth's motion through the dark matter halo of the Milky Way, and the nucleus's velocity was taken to be zero. In the frame of an observer at rest with respect to the center of mass of the dark matter-nucleus system, both will appear to be moving in opposite directions in the initial state.

We can derive the relationship between the lab velocity of the dark matter and the center-of-mass frame velocity of the dark matter. In the laboratory frame, the velocity of the dark matter with respect to the nucleus is defined by the first derivative, with respect to time, of the displacement between the dark matter particle and the nucleus. Let us denote this as $\Delta(t)$. In the center-of-mass frame, the velocity of the dark matter particle will be defined by

the time derivative of the displacement between the dark matter particle and the center of mass position, which I will denote $x(t)$. Let us place the center of mass at the origin of our coordinate system. In terms of these other displacements, the position of the center of mass along the x-axis will be given by:

$$0 = \frac{m_\chi x + m_N(\Delta - x)}{m_\chi + m_N}. \tag{39}$$

Thus

$$0 \;=\; m_\chi x + m_N(\Delta - x) \tag{40}$$

$$m_N(\Delta - x) \;=\; m_\chi x \tag{41}$$

$$m_N \Delta \;=\; (m_\chi + m_N)x. \tag{42}$$

We can now relate the velocities by taking the time derivative of this equation:

$$\frac{d}{dt} m_N \Delta \;=\; \frac{d}{dt}(m_\chi + m_N)x \tag{43}$$

$$m_N \dot{\Delta} \;=\; (m_\chi + m_N)\dot{x} \tag{44}$$

$$m_N v \;=\; (m_\chi + m_N)v_{com} \tag{45}$$

$$v_{com} \;=\; \frac{m_N}{m_\chi + m_N} v, \tag{46}$$

where $v$ is the velocity of the dark matter in the lab frame $V = 220$ km/s. You can also show that the velocity of the nucleus ($u$) relative to the center of mass will be:

$$u = \frac{m_\chi}{m_\chi + m_N} v. \tag{47}$$

Let us begin by calculating the recoil energy of the nucleus in the center-of-mass frame (Fig. 3) of the WIMP-nucleus system. Let $p$ label the three-momentum of the WIMP and $k$ label the three-momentum of the nucleus, now in the center-of-mass frame and not the laboratory frame. Then

$$\vec{p} \;=\; -\vec{k} \;\text{ (initial momentum)} \tag{48}$$

$$\vec{p'} \;=\; -\vec{k'} = \vec{q} + \mu \vec{v}_\chi \;\text{ (final momentum)}, \tag{49}$$

where the WIMP-nucleus reduced mass, $\mu$, was defined earlier as:

$$\mu = \frac{m_\chi m_N}{m_\chi + m_N}, \tag{50}$$

$v_\chi$ is the mean WIMP velocity *relative* to the target nucleus and $q$ is the momentum transfer between the WIMP and the nucleus, $\vec{q} \equiv \vec{p'} - \vec{p}$. In the center-of-mass frame and for elastic scattering, $|\vec{p}| = |\vec{p'}|$. We can then write:

$$\frac{q^2}{2} = \frac{1}{2}(\vec{p'} - \vec{p})^2 = p^2 - \vec{p} \cdot \vec{p'} = p^2(1 - \cos\theta) = \mu^2 v_\chi^2(1 - \cos\theta). \tag{51}$$

The angle, $\theta$, is as defined in Fig. 3. Often, this angle is referred to as the *recoil angle* and is denoted equivalently by $\theta_R$. To remind us this is about calculating the properties of the recoiling nucleus, which is what we hope to detect, let us adopt the convention that $\theta \to \theta_R$. Because all the initial-state action happens along the same x-axis in the laboratory or center-of-mass frames, the angle $\theta$ in this frame is identical to the angle $\theta_R$ in the laboratory frame.

The nuclear recoil (NR) energy will then be given by

$$E_R = \frac{q^2}{2m_N} = \frac{\mu^2 v_\chi^2}{m_N}(1 - \cos\theta_R). \tag{52}$$

Using this, we can then calculate the minimum dark matter particle velocity for which we expect a recoil. This will happen under the limiting case $\cos\theta_R = -1$. Then

$$E_R = \frac{\mu^2 v_\chi^2}{m_N}(1 - \cos\theta) \tag{53}$$

$$= \frac{\mu^2 v_\chi^2}{m_N}(1 - (-1)) \tag{54}$$

$$= 2\frac{\mu^2 v_\chi^2}{m_N} \tag{55}$$

$$\longrightarrow \quad v_{min} = \sqrt{\frac{m_N E_R}{2\mu^2}} = \frac{q}{2\mu}. \tag{56}$$

This relationship between the minimum speed that generates a recoil and the other properties of the system has implications:

- Lighter dark matter particles ($m_\chi \ll m_N$) must have larger minimum (threshold) velocities to generate a recoil;

- Inelastic scattering, where the nucleus is fissioned or where additional particles are generated during the scattering process, can further increase the minimal velocity needed (e.g. by the addition of other mass terms in the final state that further alters $\mu$).

Consider the average momentum transfer in an elastic scattering between a WIMP-nucleus. Let's assume a 10 GeV/c$^2$ WIMP whose speed is $\approx 100$ km/s.

$$p = m_\chi v = (10 \times 10^9 \text{ eVc}^{-2})(100 \times 10^3 \text{ ms}^{-1})\frac{c}{3 \times 10^8 \text{ ms}^{-1}} \approx 3 \text{ MeV}. \tag{57}$$

If we increase the mass of the dark matter by a factor of ten, we then also would increase the energy transferred to the nucleus to $\approx 30$ MeV.

We can also consider the de Broglie wavelength that corresponds to the momentum transfer of $\sim 10$ MeV. This is given by

$$\lambda = \frac{hc}{pc} = \frac{1239 \times 10^{-6} \text{ eV} \cdot \text{m}}{10 \times 10^6 ! \text{eV}} \sim 12 \text{ pm}. \tag{58}$$

Typical nuclear size scales go like $r \approx (1.25 \text{ fm})A^{1/3}$, where $A$ is the atomic mass number. A Uranium nucleus, for example, would have a radius of about 8 fm. The de Broglie wavelength of such a momentum transfer exceeds the size of even large nuclei, which thus implies that any such elastic scattering will be *coherent scattering*, scattering off of the entirety of the nucleus, as if the nucleus were a singular object and not a collection of nucleons, nor at a deeper level a collection of quarks and gluons.

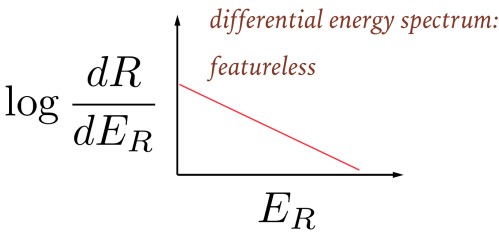

Figure 4: An example of the nucleus recoil energy spectrum from elastic WIMP-nuclei scattering.

### 3.3 Scattering Rates in a Detector - A Simple Picture

Let us now take a simplified picture of elastic scattering rates following the calculations of Lewin and Smith [9] in an imagined detector composed of atoms, where the primary target of interest is the nucleus of the atom. The differential event rate for simplified WIMP interaction (a detector stationary in the galaxy) is given by:

$$\frac{dR}{dE_R} = \frac{R_0}{E_0 r} e^{-(E_R/E_0)r} ,$$ (59)

where $R$ is the rate of elastic scattering events, $E_R$ is the recoil energy of the nucleus, $R_0$ is the total rate of such events and $E_0$ is the most probable incident energy of a WIMP (e.g. given the Maxwell-Boltzmann distribution of galactic dark matter). $r$ is a kinematic factor defined by

$$r = \frac{4 m_\chi m_N}{(m_\chi + m_N)^2}.$$ (60)

The total event rate is recovered by

$$\int_0^\infty \frac{dR}{dE_R} dE_R = R_0$$ (61)

and the mean recoiling energy is given by

$$\langle E_R \rangle = \int_0^\infty E_R \frac{dR}{dE_R} dE_R = E_0 r.$$ (62)

The spectrum of recoil energy for a given WIMP mass and WIMP velocity distribution is shown in Fig. 4. Note that this spectrum is featureless. While changing the hypothetical WIMP mass or kinematics, or the target nucleus mass, will alter the spectrum, it will always be straight and featureless (albeit with altered slope, etc.) on a logarithmic scale like this one.

We can now compute the mean nuclear recoil energy deposited in a detector. Let us assume that we have designed the detector such that $m_\chi \approx m_N = 100$ GeV/c$^2$. Then

$$\langle E_R \rangle = E_0 r = \left( \frac{1}{2} m_\chi v^2 \right) \left( \frac{4 m_\chi m_N}{(m_\chi + m_N)^2} \right)$$ (63)

since for our particular case of equivalent or near-equivalent masses, $r = 1$. If we assume that the WIMP halo is stationary and that the Sun-Earth system is passing through it at our rotation speed around the center of the galaxy, then $v \approx 220$ kms$^{-1} = 0.75 \times 10^{-3}$ $c$. If we substitute this into our equation, we obtain:

$$\langle E_R \rangle \approx 30 \text{ keV}.$$ (64)

## 3.4 Scattering Rates in a Detector - A More Complex Picture

Scattering will, of course, be more complex than presented in the previous section. We need to take into account the following considerations. First, DM will have a certain velocity distribution $f(v)$. Second, the detector is on Earth which moves around the Sun while the Sun also moves around the Galactic Center. Next, the cross-section depends upon the spin-dependence of the underlying interaction. In the simplest cases, this is either spin-independent (SI) or spin-dependent (SD).

In addition, the dark matter will scatter on nuclei that have finite size. As such, we have to consider form-factor corrections which are different for SI and SD interactions. We must also take into account that the recoil energy is not necessarily the observed energy. The detection efficiency in real life is not 100% because real detectors have certain energy resolution and energy thresholds, as well as physical processes that don't necessarily translate into pure conversion of recoil energy to some detectable form.

The total number of dark matter particles, $N$, is a product of the dark matter flux (particles per unit area per unit time), the effective area of the target and the exposure time of the experiment. There are two obvious way to increase the number of dark matter particles on target: increase the effective target area and/or operate the experiment longer. The formula is

$$N = t n v N_T \sigma, \tag{65}$$

where $n$ is the dark matter number density, $v$ the speed of the particles, $N_T$ the number of target particles, and $\sigma$ the interaction cross section of dark matter with the target particles. We will need to determine the spectrum of dark matter recoils which is the energy dependence of the number of detected dark matter particles. We do this simply by taking the derivative of Eqn (65) with respect to the recoil energy of the target particles,

$$\frac{dN}{dE_R} = t n v N_T \frac{d\sigma}{dE_R}. \tag{66}$$

We need to consider the dark matter particles are described by their local velocity distribution, $f(\vec{v})$, where $\vec{v}$ is the velocity of the dark matter particles in the reference frame of the detector (the so-called"laboratory frame"). We can insert this in the above equation by use of an integral that, performed over all velocities would yield the average speed of the dark matter particles:

$$\frac{dN}{dE_R} = t n N_T \int_{v_{min}} v f(\vec{v}) \frac{d\sigma}{dE_R} d\vec{v}, \tag{67}$$

where $v_{min}$ is the minimum speed of a dark matter particle that results in recoil energy in a collision with the nucleus as defined in Eqn. 56.

We have at our disposal some useful relationships:

$$n = \frac{\rho}{m_\chi} \tag{68}$$

$$N_T = \frac{M_T}{m_N} \text{ (with } M_T \text{ the total target mass)} \tag{69}$$

$$\epsilon = t M_T \text{ (the exposure).} \tag{70}$$

We can write:

$$\frac{dN}{dE_R} = \epsilon \frac{\rho}{m_\chi m_N} \int_{v_{min}} v f(\vec{v}) \frac{d\sigma}{dE_R} d\vec{v}. \tag{71}$$

The *event rate* is defined as $R \equiv N/\epsilon$ (number of observable events per unit target mass per unit time), so the *differential event rate* (observable events per keV of recoil energy, per kilogram,

per day) is then

$$\frac{dR}{dE_R} = \frac{\rho_0}{m_\chi m_N} \int_{v_{min}}^{\infty} v f(\vec{v}) \frac{d\sigma_{\chi N}}{dE_R} d\vec{v}, \tag{72}$$

where I solidify the notation so that $\rho_0$ is the *local dark matter density* (e.g. in our region of the Milky Way), $f(v)$ is the WIMP speed distribution in the detector frame, and $\sigma_{\chi N}$ is the WIMP-nucleus scattering cross-section. To convert this latter cross section to a WIMP-nucleon cross-section requires nuclear physics to map overall nuclear structure onto individual nucleons in the nucleus.

Dark matter detection, then, is a synthesis of astrophysics (to estimate the local dark matter density), particle physics (to establish a framework for the scattering), and nuclear physics (to map the nuclear to the nucleon properties).

> **Dark Matter Detection Equations**
>
> In summary, the dark matter detection equations of note are
>
> $$E_R = \left(\frac{m_\chi m_N}{m_\chi + m_N}\right)^2 \frac{v^2}{m_N}(1 - \cos\theta_R) \tag{73}$$
>
> and
>
> $$\frac{dR}{dE_R} = \frac{\rho_0}{m_\chi m_N} \int_{v_{min}}^{\infty} v f(\vec{v}) \frac{d\sigma_{\chi N}}{dE_R} d\vec{v} \tag{74}$$

## 3.5 The Scattering Cross Section

The event rate in a potential detector is determined by integrating over all recoils.

$$R = \int_{E_{threshold}}^{\infty} dE_R \frac{\rho_0}{m_\chi m_N} \int_{v_{min}}^{\infty} v f(\vec{v}) \frac{d}{dE_R} \sigma_{\chi N}(v, E_R) d\vec{v}. \tag{75}$$

In this section, I want to focus on the elements that enter the differential cross-section, $d\sigma_{\chi N}/dE_R$. First, let's consider that the scattering process takes place in the non-relativistic limit. It can be approximated as an isotropic scattering distribution, such that

$$\frac{d\sigma}{d\cos\theta} = \text{constant} = \frac{\sigma}{2}. \tag{76}$$

The maximum kinetic energy that can be transferred to the nucleus occurs when $\cos\theta_R = -1$ (100% back-scatter of the nucleus in the center-of-mass frame), such that $E_R^{max} = 2\mu^2 v^2/m_N$. In terms of this quantity, Eqn 73 becomes

$$E_R = \frac{E_R^{max}}{2}(1 - \cos\theta_R). \tag{77}$$

We can obtain the differential of the recoil energy with respect to scattering angle:

$$\frac{dE_R}{d\cos\theta} = \frac{E_R^{max}}{2}. \tag{78}$$

From this, we can then write the differential of the cross-section with respect to recoil energy using the chain rule:

$$\frac{d\sigma}{dE_R} = \frac{d\sigma}{d\cos\theta}\frac{d\cos\theta}{dE_R} = \frac{\sigma}{2}\frac{2}{E_R^{max}} = \frac{\sigma}{E_R^{max}} = \frac{m_N}{2\mu^2}\frac{\sigma}{v^2}. \tag{79}$$

The momentum transfer involved in these collisions corresponds to a de Broglie wavelength in excess of the nuclear size. Thus, we are coherently scattering off the entire nucleus. Especially for lighter nuclei, the WIMP doesn't "see" the nucleons at all. For light nuclei, one can make a good approximation that the nucleus is nearly the sum of its parts (nucleons) without significant corrections to that approximation. For light nuclei, it's therefore possible to translate a scattering rate on the nucleus into a scattering rate on individual nucleons (e.g. to infer the WIMP-proton or WIMP-neutron cross-section).

However, it's more typical for heavy nuclei to be used in experiments. Given our early considerations, this is for obvious reasons: more target cross-sectional area means more anticipated WIMP interactions per unit mass, per unit time. Heavy nuclei receive significant corrections from the "sum of nucleons" approximation when describing the nucleus as a whole. These corrections are summarized by *nuclear form factors*, $F$. For example, let us consider that we are not certain whether the WIMP-nucleus interactions may have spin-independent or spin-dependent effects. It's natural, therefore, to assume that the cross-section in totum will be a sum of both possibilities, e.g.

$$\frac{d\sigma}{dE_R} = \left[ \left( \frac{d\sigma}{dE_R} \right)_{SI} + \left( \frac{d\sigma}{dE_R} \right)_{SD} \right], \tag{80}$$

where "SI" ("SD") denotes the spin-independent (spin-dependent) scattering. Spin-independent effects would arise from a scalar or vector dark matter candidate coupling to quarks inside the nucleons, while spin-dependent effects would arise from an axial-vector coupling to those same quarks. To perform calculations, it's straight forward to begin by assuming that these effects add coherently with corrections from the parton-level up to the nuclear scale:

$$\frac{d\sigma}{dE_R} = \frac{m_N}{2\mu^2 v^2} \left[ \sigma_0^{SI} F_{SI}^2 + \sigma_0^{SD} F_{SD}^2 \right]. \tag{81}$$

The form factors encode the dependence on the momentum transfer and the nuclear structure without making explicit the details of these dependencies; the cross-sections encode the parton-level (e.g. quark-level) particle physics of the WIMP-parton interaction (e.g. Feynman diagrams describing the leading interactions possible between fundamental particles).

The Form Factors are an area of rather active research. Nuclear physics, to say the least, is complicated. The nature of quantum chromodynamics and the energy domain of the nucleus makes high-precision, direct analytical calculations of such form factors nearly impossible. Instead, a synthesis of nuclear scattering data and computational approaches are typically used to constrain or infer these corrections. It is also possible to make models of these form factors based on theoretical or data-driven considerations.

One example is the Helm Form Factor [10] [11], which applies to spin-independent interactions,

$$F(q) = \left( \frac{3 j_1(qR_1)}{qR_1} \right)^2 \exp(-q^2 s^2 / 2). \tag{82}$$

Here, $j_1$ is the Spherical Bessel Function,

$$j_1(x) = \frac{\sin(x)}{x^2} - \frac{\cos(x)}{x}. \tag{83}$$

$q$ is the momentum transfer, $s$ is the nuclear skin thickness ($s \sim 1$ fm), and $R_1$ is the effective nuclear radius, which should be approximated as $R_1 = 1.25$ fm $\times A^{1/3}$, where $A$ is the atomic mass number.

For spin-dependent interactions, we can write

$$F^2(E_R) = \frac{S(E_R)}{S(0)} \tag{84}$$

| Nucleus | $Z$ | Odd Nucleon | $J$ | $\langle S_p \rangle$ | $\langle S_n \rangle$ | $C_A^p/C_p$ | $C_A^n/C_n$ |
|---------|-----|-------------|-----|-----------------------|-----------------------|-------------|-------------|
| $^{19}$F | 9 | p | 1/2 | 0.441 | $-0.109$ | $7.78 \times 10^{-1}$ | $4.75 \times 10^{-2}$ |
| $^{23}$Na | 11 | p | 3/2 | 0.248 | 0.020 | $1.37 \times 10^{-1}$ | $8.89 \times 10^{-4}$ |
| $^{27}$Al | 13 | p | 5/2 | $-0.343$ | 0.030 | $2.20 \times 10^{-1}$ | $1.68 \times 10^{-3}$ |
| $^{29}$Si | 14 | n | 1/2 | $-0.002$ | 0.130 | $1.60 \times 10^{-5}$ | $6.76 \times 10^{-2}$ |
| $^{35}$Cl | 17 | p | 3/2 | $-0.083$ | 0.004 | $1.53 \times 10^{-2}$ | $3.56 \times 10^{-5}$ |
| $^{39}$K | 19 | p | 3/2 | $-0.180$ | 0.050 | $7.20 \times 10^{-2}$ | $5.56 \times 10^{-3}$ |
| $^{73}$Ge | 32 | n | 9/2 | 0.030 | 0.378 | $1.47 \times 10^{-3}$ | $2.33 \times 10^{-1}$ |
| $^{93}$Nb | 41 | p | 9/2 | 0.460 | 0.080 | $3.45 \times 10^{-1}$ | $1.04 \times 10^{-2}$ |
| $^{125}$Te | 52 | n | 1/2 | 0.001 | 0.287 | $4.00 \times 10^{-6}$ | $3.29 \times 10^{-1}$ |
| $^{127}$I | 53 | p | 5/2 | 0.309 | 0.075 | $1.78 \times 10^{-1}$ | $1.05 \times 10^{-2}$ |
| $^{129}$Xe | 54 | n | 1/2 | 0.028 | 0.359 | $3.14 \times 10^{-3}$ | $5.16 \times 10^{-1}$ |
| $^{131}$Xe | 54 | n | 3/2 | $-0.009$ | $-0.227$ | $1.80 \times 10^{-4}$ | $1.15 \times 10^{-1}$ |

Figure 5: From Ref. [13], this table provides values for terms that appear in Eqn. 88. The quantities $C_A^{p,n}$ are defined as $C_A^i \equiv (8/\pi)(a_i \langle S_i \rangle)^2 [(J+1)/J]$.

where

$$S(E_R) = a_0^2 S_{00}(E_R) a_1^2 S_{11}(E_R) + a_0 a_1 2 S_{01}(E_R) \tag{85}$$

and

$$a_0 = a_p + a_n, \, a_1 = a_p - a_n. \tag{86}$$

Here, $S_{ij}$ are the isoscalar (0), isovector (1), and interference form factors while $a_i$ are the isoscalar or isovector coupling constants.

The spin-dependent form factor is the superposition of form-factor components (isoscalar, etc.) normalized to that superposition at zero recoil energy, the case that no energy is transferred to the nucleus.

The forms of these SI and SD cross-sections that most often appear in the literature are

**Spin-Independent and Spin-Dependent Cross Sections**

$$\sigma_0^{SI} = \frac{4\mu^2}{\pi} \left[ Z f_p + (A - Z) f_n \right]^2 \propto A^2 \tag{87}$$

$$\sigma_0^{SD} = \frac{32 G_F^2 \mu^2}{\pi} \frac{J+1}{J} \left[ a_p \langle S_p \rangle + a_n \langle S_n \rangle \right]^2 \tag{88}$$

Eqn. 87 is valid for spin-independent interactions only. Furthermore, we generally assume a low momentum transfer which corresponds to low $q^2$. As such, the scattering process cannot resolve the difference between the proton and neutron at the parton level. In that case, the four-fermion coupling factors satisfy $f_p \sim f_n$ [12] (This is a model-dependent assumption that may need to be revisited if WIMP scattering is observed). The scattering is overall simply proportional to the atomic mass number (larger $A$ means more interactions) and adds coherently with an $A^2$ enhancement.

However, in Eqn. 88 the nuclear angular momentum, $J$, explicitly appears, as do the individual couplings to the proton and neutron ($a_i$) and the expectation values of the nucleon spins $\langle S_i \rangle$. Different nuclei can lead to very different scattering rates, even for nearly identical values of $A$; in fact, isotopic variations among elements will potentially significantly shift the scattering process. An example of how to factor such considerations into the choice of a particular target element and isotope is shown in Fig. 5. This table from Ref [13] summarizes efforts that have gone into modeling the nuclear physics considerations which influence experimental choices and design.

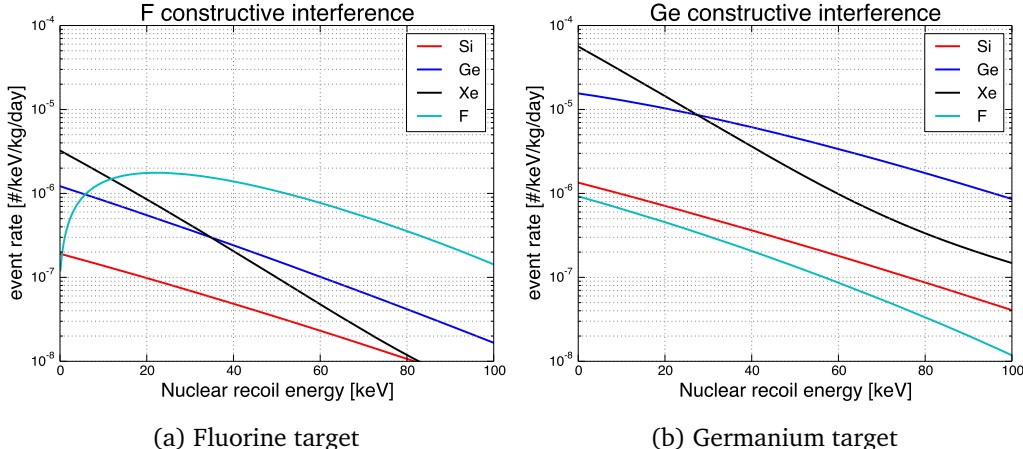

| (a) Fluorine target | (b) Germanium target |

Figure 6: Figures from Ref. [17]. Differential event rate evaluated at the $\mathcal{O}_8/\mathcal{O}_9$ constructive interference vector from Fluorine and Germanium targets.

### 3.6 Effective Field Theory and the Complexity of Dark Matter Scattering

The treatment of scattering so far has been simple and generic, devoid of a particular underlying theoretical framework except to assume basic non-relativistic quantum mechanics. In reality, we know that these more approximate models are rooted in quantum field theories. We do not yet know what field theory framework provides the proper description of dark matter, so it is impossible to be explicit about scattering calculations.

However, effective quantum field theories (EFTs) have proven to be a useful tool for establishing all possible kinds of interactions that can take place between fermions and bosons. As measurements are made with experiments, it may be possible to eliminate certain interaction terms from consideration; some may be ruled out a priori as they provide undesirable or incompatible features for dark matter particles. Such effective field theories usually begin by considering leading-order and next-to-leading-order contributions to interactions, where the "order" here refers to the cut-off in a perturbation expansion of the theory that is usually conducted in terms of the coupling strength(s) of the interaction(s) involved in the theory.

Such an EFT would contain 14 operators that rely on a range of nuclear properties in addition to the SI and SD cases [14] [15] [16] [17]. They can be combined and grouped such that the WIMP-nucleon cross section depends on six independent nuclear response functions: (a) one "spin-independent, (b) two "spin-dependent," (c) and three "velocity-dependent" functions. Two pairs of these interfere, resulting in eight independent parameters that can be probed by comparing EFT predictions with real experimental data (e.g. scattering observations or non-observations). Each operator connects an initial state and a final state, and in effect determines the nature and outcome of each kind of interaction.

Basic matrix element considerations apply here. Consider some generic operator, $\mathcal{O}$, that describes an interaction that takes some initial state $|i\rangle$ to some final state $|f\rangle$. The matrix element that describes the square-root of the probability density for this transition via this operator is

$$\mathcal{M} = \langle i | \mathcal{O} | f \rangle. \tag{89}$$

The probability of this transition is then given by (or is proportional to)

$$|\mathcal{M}|^2 = |\langle i | \mathcal{O} | f \rangle|^2. \tag{90}$$

The 14 operators in the EFT used to describe WIMP-nucleon interactions are

$$\mathcal{O}_1 = 1_\chi 1_N \tag{91}$$

$$\mathcal{O}_3 = i\vec{S}_N \cdot \left[ \frac{\vec{q}}{m_N} \times \vec{v}^\perp \right] \tag{92}$$

$$\mathcal{O}_4 = \vec{S}_\chi \cdot \vec{S}_N \tag{93}$$

$$\mathcal{O}_5 = i\vec{S}_\chi \cdot \left[ \frac{\vec{q}}{m_N} \times \vec{v}^\perp \right] \tag{94}$$

$$\mathcal{O}_6 = \left[ \vec{S}_\chi \cdot \frac{\vec{q}}{m_N} \right] \cdot \left[ \vec{S}_N \cdot \frac{\vec{q}}{m_N} \right] \tag{95}$$

$$\mathcal{O}_7 = \vec{S}_N \cdot \vec{v}^\perp \tag{96}$$

$$\mathcal{O}_8 = \vec{S}_\chi \cdot \vec{v}^\perp \tag{97}$$

$$\mathcal{O}_9 = i\vec{S}_\chi \cdot \left[ \vec{S}_N \times \frac{\vec{q}}{m_N} \right] \tag{98}$$

$$\mathcal{O}_{10} = i\vec{S}_N \cdot \frac{\vec{q}}{m_N} \tag{99}$$

$$\mathcal{O}_{11} = i\vec{S}_\chi \cdot \frac{\vec{q}}{m_N} \tag{100}$$

$$\mathcal{O}_{12} = \vec{S}_\chi \cdot \left[ \vec{S}_N \times \vec{v}^\perp \right] \tag{101}$$

$$\mathcal{O}_{13} = i\left[ \vec{S}_\chi \cdot \vec{v}^\perp \right] \left[ \vec{S}_N \cdot \frac{\vec{q}}{m_N} \right] \tag{102}$$

$$\mathcal{O}_{14} = i\left[ \vec{S}_\chi \cdot \frac{\vec{q}}{m_N} \right] \left[ \vec{S}_N \cdot \vec{v}^\perp \right] \tag{103}$$

$$\mathcal{O}_{15} = -\left[ \vec{S}_\chi \cdot \frac{\vec{q}}{m_N} \right] \left[ (\vec{S}_N \times \vec{v}^\perp) \cdot \frac{\vec{q}}{m_N} \right], \tag{104}$$

where $\vec{v}^\perp$ is the relative velocity between the incoming WIMP and the nucleon, $q$ is the momentum transfer, $\vec{S}_\chi$ is the WIMP spin, and $\vec{S}_N$ is the nucleon spin. In addition, each operator can independently couple to protons and neutrons. You will notice that $\mathcal{O}_2$ is missing from the list; this is because it cannot arise in the non-relativistic limit, which we are considering here.

As a result of this complex web of operators, representing different spin-independent, spin-dependent, and velocity-dependent interactions, WIMP scattering can look very different across a range of common or plausible target materials. For example, nuclear responses for different target elements vary, some EFT operations have momentum dependence, and EFT operators can interfere. An example of these effects is shown in Fig. 6. This illustrates differences evaluating event rates using the $\mathcal{O}_8$ and $\mathcal{O}_9$ constructive interference vector. This effect results in different rates between targets as well as different recoil energy spectral shapes. Not all targets are equal, nor will two different targets tells you the same thing about the properties of the dark matter or their interactions with the nucleus.

A robust dark matter direct detection program with different target materials will be needed to identify which operators are contributing to any detected signal. A rich and diverse search program will need multiple targets to map out the physics of WIMP-nucleon interactions.

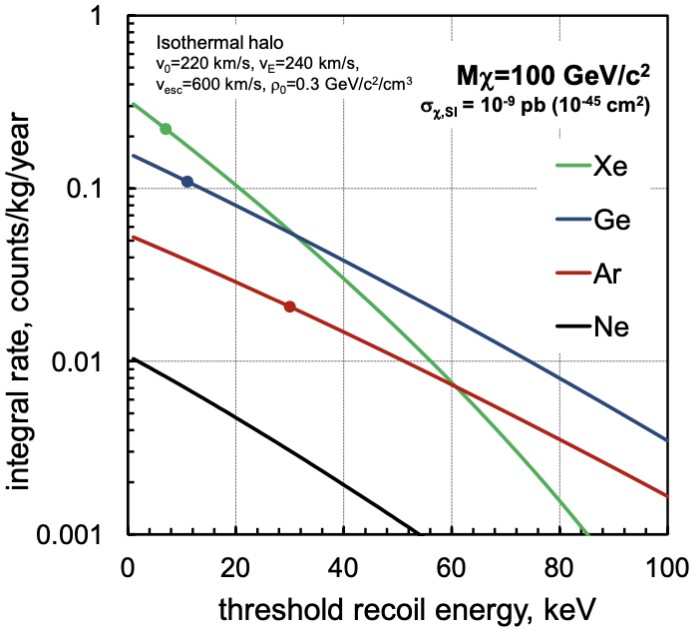

Figure 7: Event rates in units of counts per 10 kg of material per year of data-taking vs. energy threshold in keV for different target materials. Taken from [18]

## 3.7 The WIMP Detection Challenge

Another challenge that comes from basic or EFT-driven scattering considerations is the overall low rate at which we can expect nuclear recoils to occur. For example, event rates in different materials as a function of energy threshold for a 100 GeV/c$^2$ WIMP and a WIMP-nucleon total cross section of $10^{-45}$ cm$^2$ (compatible with upper bounds placed by leading experiments in the field today), are illustrated in Fig. 7 [18]. You will notice that we only expect a few events per kg-year of exposure.

Elastic scattering of a WIMP deposits small amounts of energy into a recoiling nucleus, amounting to just a few tens of keV. The energy spectrum of the recoil is featureless and falls exponentially as a function of the recoil energy. There are not expected to be any resonances, peaks, knees, or breaks in this spectrum, making it hard to distinguish from other processes with similar featureless energy spectra. The overall event rate is extremely low, requiring large targets and long periods of data-taking. This alone won't be sufficient. Materials are suffused with naturally occurring radioactive backgrounds which are, even under fair conditions, capable of producing nuclear recoils at rates far higher than these.

For lower-mass dark matter candidates, the challenge for such liquid- and solid-state material approaches is even greater. The event rate vs. energy threshold for various WIMP candidate masses is shown in Fig. 8. From the earlier exploration of recoil energy in non-relativistic elastic scattering we know that as the WIMP mass declines, the challenge of generating a nuclear recoil grows larger. In addition, different target materials will respond differently to the same kind of low-mass WIMP candidate (Fig. 9). Lower WIMP masses will demand, more and more, materials and approaches that can go to lower and lower recoil energy thresholds.

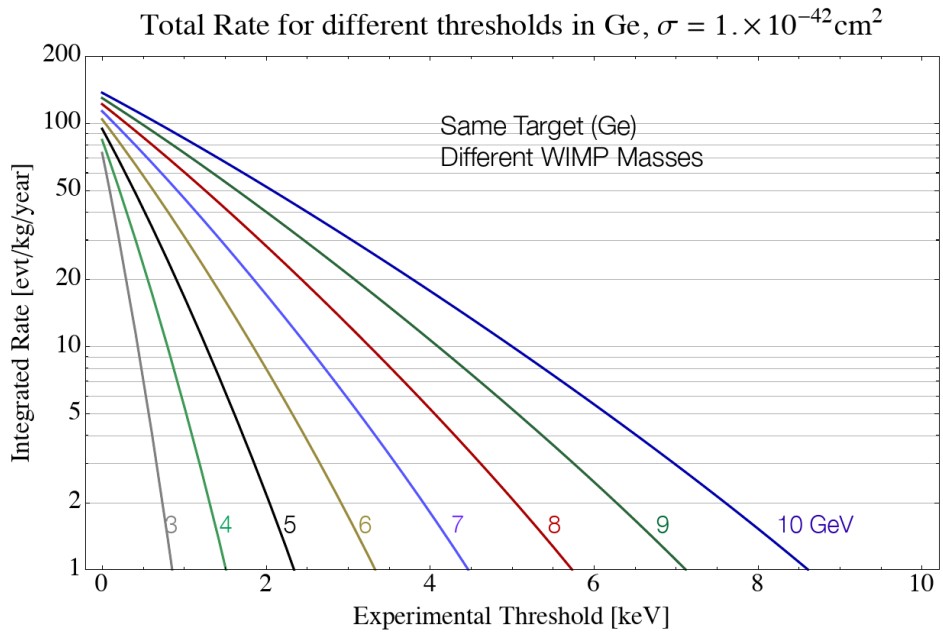

Figure 8: Event rates in units of counts per kg of material per year of data-taking vs. energy threshold (in keV) for different low WIMP masses. The target material is fixed on Germanium. Figure courtesy of Enectali Figueroa-Feliciano.

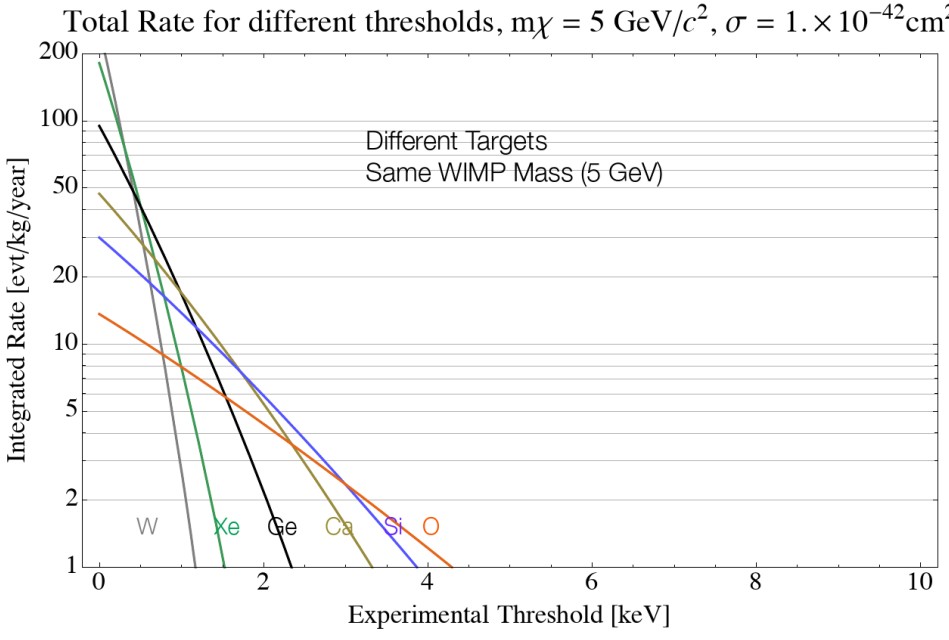

Figure 9: Event rates in units of counts per kg of material per year of data-taking vs. energy threshold in keV for different materials. The WIMP mass is fixed at 5 GeV/$c^2$. Figure courtesy of Enectali Figueroa-Feliciano.

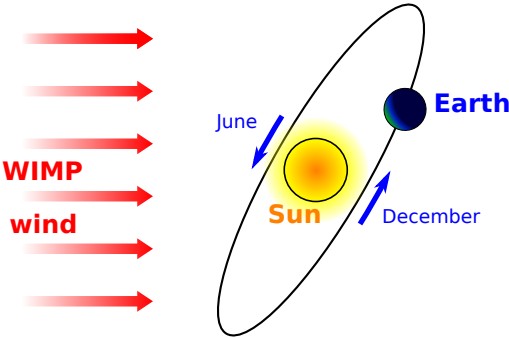

Figure 10: A schematic showing the relationship of the Earth's orbit around the Sun to the direction of travel of the Sun through the Milky Way. The WIMP "wind" points opposite that direction of travel. Figure from Ref. [19].

## 3.8 Time-Dependence of the Scattering Rate

The focus has been on the event rate integrated over time, but not on the potential time structure of the rate itself. There is a strong reason to believe that the interaction rate of dark matter on a target will be time dependant. The most obvious reason for this is that the average speed of the dark matter is determined by the estimated speed with which the Sun orbits the center of the Milky Way. However, the Earth orbits the Sun, so that half the year our planet is headed "into the WIMP wind" while in the other half of the year we are headed "out of the WIMP wind." These tiny expected variations in the average speed of WIMPs could manifest in a detector as an *annual modulation* of the event rate.

The motion of the Sun through the WIMP halo is in the direction of the constellation Cygnus (Fig. 10) and is in the plane of the galaxy. However, the plane of the Earth's orbit is not aligned with that direction, but rather tilted about 60° above that line of motion. We should only consider the change in the WIMP wind based on the component of the Earth's orbital velocity that is parallel to the Sun-Cygnus line.

Earth's speed with respect to the galactic rest frame, taken also to be the WIMP halo rest frame, is largest in the summer when the components of Earth's orbital velocity in the direction of solar motion is largest. The number of WIMPs with high (low) speeds in the detector rest frame is largest (smallest) in the summer. As a result, in the Northern Hemisphere, we would expect any annual modulation of the event rate to peak during the summer months and be minimal in the winter months.

We can estimate the degree of this effect [19]. The Earth's speed around the Sun is much smaller than the speed of the Sun about the center of the Milky Way. In numbers, $v_{orbit}/v_{sun} \approx 0.07$. This small value of the relative speeds provides an opportunity to Taylor expand the differential event rate to make a leading order approximation of the effect.

$$\frac{dR}{dE_R}(E_R, t) \approx \frac{dR}{dE_R}\left[1 + \Delta(E_R)\cos\frac{2\pi(t - t_0)}{T}\right], \tag{105}$$

where $T$ is the period of the modulation and is expected to be one Earth orbital period, or 365.25 days, and $t_0$ is the phase which, due to the relationship between the plane of orbit and the motion of the Sun is $t_0 = 150$ days. The factor $\Delta_{E_R}$ is the *modulation amplitude*. To detect the seasonal variation due to the changing recoil energy spectrum across the year, one needs sufficient detections to tell the difference between the higher number in the Northern summer and the lower number in the Northern winter. This requires at least the observation of 1000 interactions to detect what is expected to be a few percent effect. This effect on an example generic recoil energy spectrum is shown in Fig. 11.

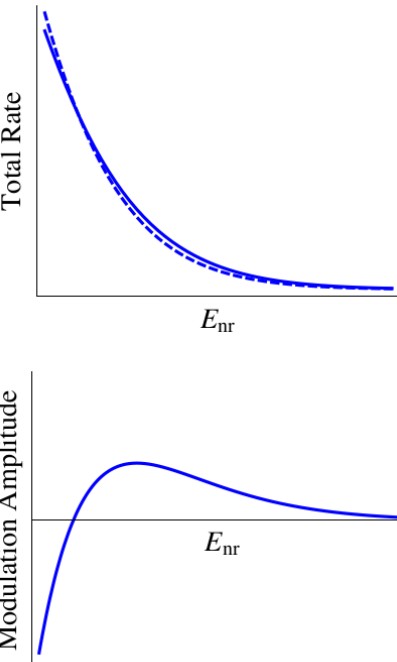

Figure 11: The difference in the recoil energy spectrum (top) comparing the maximum (solid line) and minimum (dashed line) times of the year for an annual modulation of WIMP interactions. The bottom graph shows the ratio of the modulation amplitude for those two cases. Figure excerpted from Ref. [19].

In addition to time dependence, if detectors can be made sensitive to the direction of the nuclear recoil this can help to understand the modulation signal. Directionality [20] can also be used for a variety of other purposes, including the suppression of confounding processes in detector systems. The Earth's motion with respect to the Galactic rest frame produces a direction dependence of the recoil spectrum. The peak WIMP flux will appear to point from the direction of solar motion, from Cygnus toward us. Assuming a smooth and isotropic WIMP halo, the recoil rate is peaked in the opposite direction of our motion toward Cygnus. In the lab frame, this direction varies over the course of a day due to Earth's rotation as illustrated in Fig. 12. In addition, due to the fact that the direction to Cygnus varies by hemisphere, detectors places on different halves of the Earth should see slightly different directional effects. This is another example of the importance of multiple, independent, and complementary dark matter instruments.

As a result of this directional dependence, we should actually expect the number of nuclear recoils along a particular direction in the lab frame to change over the course of a single day. A daily modulation in directionality could be visible with a sensitive enough detector. Assuming the Standard Halo model, the dependence is given by

$$\frac{dR}{dE_R \cos\gamma} = \frac{\rho_0 \sigma_{\chi N}}{\sqrt{\pi}\sigma_v} \frac{m_N}{2m_\chi \mu^2} \exp\left[-\frac{[(v_{orbit}^E v_{sun})\cos\gamma - v_{min}]^2}{\sigma_v^2}\right], \tag{106}$$

where $v_{orbit}^E$ is the component of the Earth's velocity around the Sun parallel to the Sun-Cygnus line and $\gamma$ is the angle between the observed nuclear recoil and the Sun-Cygnus line. The event rate in the forward direction is expected to be up to an order of magnitude larger than backward direction [22]. A detector measuring the axis and direction of the recoil with good angular resolution needs only a few tens of events to distinguish dark matter interactions from

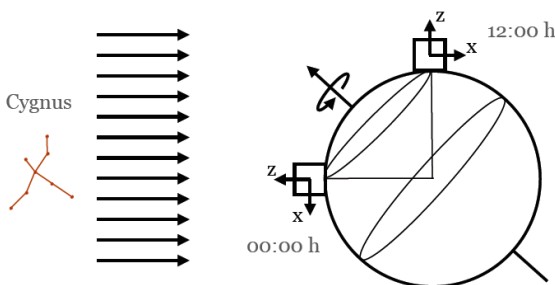

Figure 12: Spin axis of Earth with respect to the WIMP "wind" generated by the motion of the Earth through the Galaxy. Figure excerpted from Ref. [21]

isotropic backgrounds that have no preferred direction (e.g. not aligned with the Sun-Cygnus line).

## 4 Backgrounds to Direct Detection Methodologies

There are many known and potential sources of confounding detector signatures - *background processes* or *backgrounds* for short. The most prominent of these is environmental radioactivity. This category includes airborne radon and its daughters as well as radio-impurities in materials used for the detector and shielding construction. One of the worst potential backgrounds connected to environmental radioactivity is radiogenic neutrons with energies below 10 MeV (neutrons emitted during nuclear decay). These are sourced by alpha radiation interactions with neutrons and fission reactions. Then there are the cosmogenic backgrounds, such as cosmic ray muons passing through the instrumentation or, worse, spallation neutrons produced as a secondary by-product of cosmic rays which then enter the detector volume. Materials themselves, later used in construction, become activated due to natural exposure to cosmic rays or radionuclides near Earth's surface. Neutrinos from a wide variety of sources, from terrestrial origins to solar and cosmic neutrinos, offer a weakly interacting low-mass background that can interact with electrons or the nucleus. Of course, there is always the worst background of all: the one no one has thought of yet.

### 4.1 Cosmic Rays

Let's tackle cosmic rays immediately. The intensity of cosmic rays increases with altitude and decreases as one goes below the surface of the Earth. For this reason direct detection dark matter experiments seek laboratories situated beneath the surface of the Earth, either in specially constructed shallow underground sites or in deep mines that are also home to laboratory facilities. An overburden of rock will protect an experiment from the leading problem, the direct exposure to cosmic rays. However, it will then present the secondary problem: backgrounds generated by cosmic ray interactions in the overburden material. As shown in Fig. 13 these secondary particles can be reduced by increasing further the depth of the laboratory housing the dark matter experiment.

Many experiments also employ an *active cosmic ray veto system*, a detector subsystem designed to identify moments when a cosmic ray primary or secondary has entered the volume of the main detector system. The time information about these events can then be used to veto from consideration any activity in the primary detector system.

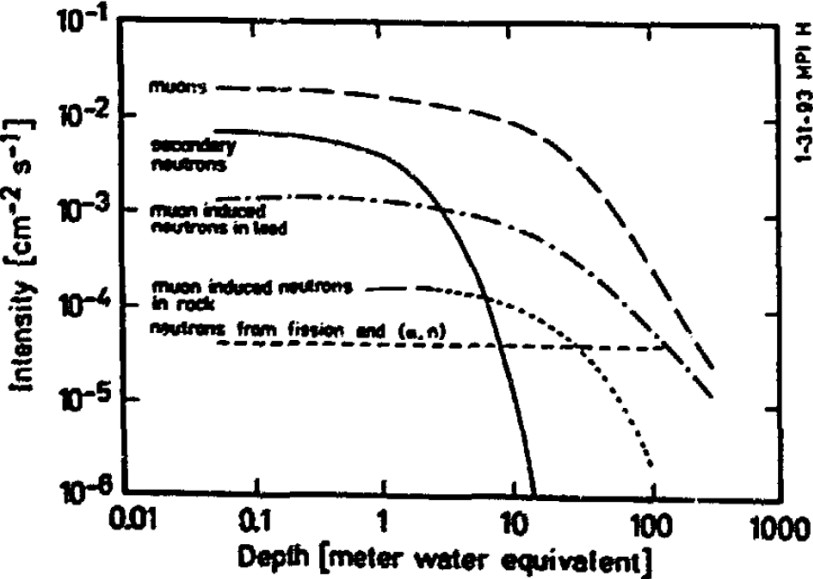

Figure 13: The cosmic ray primary intensity and the intensity of production of secondary particles by cosmic ray primaries. This is shown as a function of depth below the surface of the Earth, where depth is calibrated to meters-water-equivalent (the equivalent depth if only water composed the overburden). Figure excerpted from Ref. [23].

## 4.2 Radioactivity Primer

It is useful to remind ourselves of some basic features of radioactive decay. Fundamentally, the mathematics of this process is simple. Radioactive decay is a spontaneous process. The rate of change $-dN/dt$ in the present population of particles $N$ is proportional to the size of that population:

$$-\frac{dN}{dt} = \lambda N. \tag{107}$$

The *activity* of a sample $A$ is defined as this rate of change of the population, $A \equiv dN/dt$. Here, $\lambda$ is referred to as the decay constant (potentially quite different for each radioisotope or new unstable species). One can find a solution to this equation rather straight-forwardly: we see a function, $N(t)$, such that the first derivative of this function with respect to time returns the function itself multiplied by a constant, $\lambda$. This is a classical hallmark of the exponential function. We guess that

$$N(t) = Ce^{Dt}. \tag{108}$$

We need to solve for $C$ and $D$. To solve for $C$, consider the boundary condition $t = 0$:

$$N(0) = Ce^{D \cdot 0} = C. \tag{109}$$

We can identify $C$ as the size of the population at $t = 0$, which I denote $N_0$. To identify $D$, take the derivative of the solution:

$$\frac{d}{dt}N(t) = \frac{d}{dt}N_0 d^{Dt} = N_0 De^{Dt} = DN(t) = -\lambda N(t). \tag{110}$$

We identify $D = -\lambda$. Thus our solution is of the form:

$$N(t) = N_0 e^{-\lambda t}. \tag{111}$$

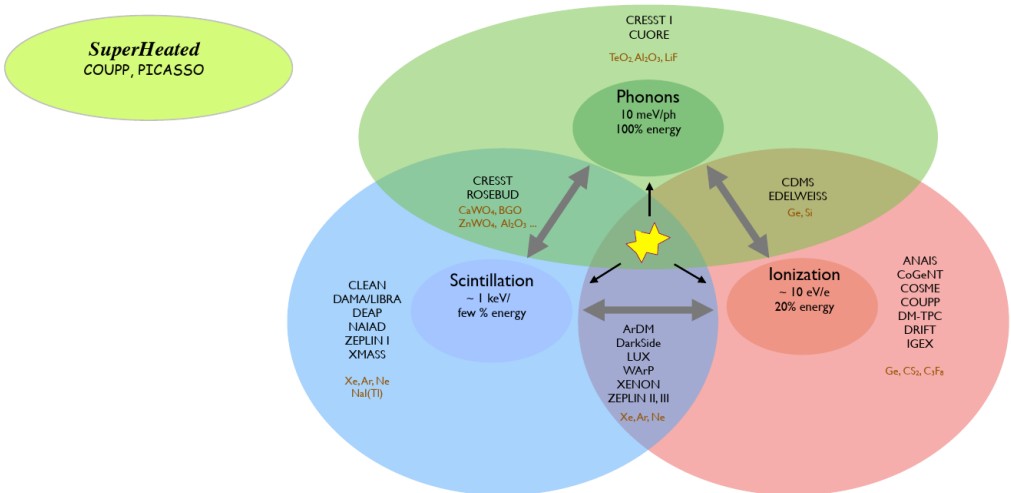

Figure 14: Signatures for detecting electron and nuclear recoils in materials and experiments that employ them individually or in combination.

$\lambda$ has units of $s^{-1}$. As such, it can be identified with a time constant, $\lambda = 1/\tau$. With a little more work we can see that the decay time of the population, $\tau$, is the time such that when $t = \tau$, 63.2% of the original population has decayed. This can be related to the *half life* of the population - the time $t_{1/2}$ at which 50% of the population has decayed - by the equation:

$$\lambda = \frac{\ln(2)}{t_{1/2}} \longrightarrow \tau = \frac{t_{1/2}}{\ln(2)}. \tag{112}$$

The initial number of radioisotopes in a population can be determined by

$$N_0 = \frac{N_A}{m_{isotope}} \times M_{isotope}, \tag{113}$$

where $N_A$ is Avogadro's Number, $m_{isotope}$ is the atomic mass of the radionuclide, and $M_{isotope}$ is the total mass of the the radionuclide. There is also the concept of *abundance* of an isotope, which refers to the total amount of that isotope present relative to the stable isotopes in a given sample of material.

## 4.3 Event Signatures

When radiation emitted by an unstable isotope impinges on a detector medium, what may be the signatures of these interactions? We would expect two kinds of signature: interactions with the atomic electrons or with the nucleus. For example, photons (e.g. gamma rays) and electrons (e.g. beta radiation) will tend to scatter off the atomic electrons. Alpha particles will ionize atomic electrons, but can also interact with the atomic nucleus. Neutrons will interact exclusively with the atomic nucleus. This latter fact is what makes them particularly pernicious: WIMPS and neutrons preferentially are expected to interact with the same part of an atom.

The interaction with atomic electrons generates what is known as an *electron recoil*. Gamma rays are the most prevalent source of such recoils, while beta radiation (fast electrons) tend to cause recoils near the surface of detector materials and due to their penetrating capabilities recoils in the *bulk* of the material. Electron recoils generate significant electron currents in materials, in addition to other phenomena, and in general can be more easily differentiated from WIMP interactions.

However, neutrons and WIMPs will both interact with the nucleus and generate the nuclear recoils discussed earlier in this chapter. Neutron interactions appear like a 1 GeV WIMP striking a nucleus with a well-defined interaction cross-section, with the rest of their effects being defined by what speed they strike the nucleus. In fact, you can apply Eqns. 73 and 74 using the neutron mass and its nuclear interaction cross-section, which is very well understood.

Electric currents are not the only response of materials to electron and nuclear recoils. In fact, materials are generally selected because they provide a range of signatures whose relative rates vary by what is interacting in the material. These signatures can be summarized broadly as follows:

- Heat

  - Phonons: for systems with a regular order (e.g. solid state materials) and restoring forces to maintain the structure, phonons which are quanta of vibrational energy, are a key signature. For these to be useful the material must be kept extremely cold to reduce or remove normal thermal noise from the system. In a crystal lattice kept at such low temperatures, a single atomic recoil whether electronic or nuclear can setup phonons in the system. Phonons can carry about 10 meV per quanta and, if detected at high efficiency, can represent 100% of the original energy deposited in the material.

  - Superheating: for systems kept in a liquid state and under immense pressure, it is possible to contrive the thermodynamics of the system such that a nuclear or electronic recoil will cause the material to boil, generating small bubbles that can be imaged. This is precisely the operating characteristic of bubble chambers, and new versions of these devices play a role in dark matter searches.

- Light

  - Scintillation: this is a key means to detect interactions in materials. Radiation absorbed by a material will result in the emission of light. With appropriate chemistry and detector instrumentation, it is possible to observe and capture this light and determine a portion of the energy present in the original interaction. However, in general scintillation only captures a few percent of the original interaction energy so even with high detection efficiency one is only sampling a small portion of the original interaction.

- Ionization

  - The removal of electrons from parent atoms creates both a population of free electrons that can then traverse the material. These electrons and absence of electron (holes) can be read out using modern electronics. Ionization lends itself to the application of an electric field in the material to amplify the signals or drift them to desirable locations for read out. Each conduction electron will carry about 10 eV and with efficient readout one has access to $\sim$20% of the original interaction energy.

A diagram representing these major signatures as well as experiments that employ each by itself, or signatures in combination, is shown in Fig. 14.

WIMPs and neutrons scatter off of nuclei in nuclear recoils (NR). Most backgrounds scatter off of electrons in electron recoils (ER). Detectors have different responses to NR and ER, allowing for potential discrimination of these different kinds of interations. However, the concept of the *quenching factor* (QF) becomes quite important in the discussion of these interactions and

detection of their signatures. The QF describes the difference in the amount of visible energy in a detector to these two classes of events (NR and ER). The ER signal will be measured in keVee (*keV electron equivalent*) while the NR signal will be measured in keVnr (*keV nuclear recoil*). These are two different energy scales that, in principle, need to be calibrated one to the other to know, in a material, how to translate between then. For NR events,

$$E_{visible}(\text{keVee}) = QF \times E_{recoil}(\text{keVnr}). \tag{114}$$

The employment of well-defined radiation sources such as a calibrated gamma or neutron emitter whose emission energies are known can be used to determine the quenching factor that relates these two energy scales. Gamma emitters will preferentially induce electron recoils, while neutron sources will induce nuclear recoils. The energy spectra of these two sets of recoils, taken in the same material and taking into account the energies of the gamma and neutron radiation, can be combined to determined the relationship between the ER and NR scales.

Experiments can be designed to measure both ER and NR signatures. Their combination is then employed to develop regions of the ER and NR signatures rich in potential WIMP signatures (e.g. low ER, high NR) or backgrounds (e.g. high ER, low NR). One example of this is a nuclear recoil in an ultra-cold germanium crystal). Such a solid-state device allows for both the production of electron-hole pairs through ionization and for the production of phonons through recoils of the lattice atoms resulting from striking the nucleus of such an atom. In germanium, we find that $E_{visible} \approx (1/3)E_{recoil}$, which means the quenching factor for germanium is $\sim 30\%$. For more details and examples of experiments that use these techniques see Sec. 7.3.

Another good example of this is a high-pressure liquid state system such as a bubble chamber. In this case, we have a volume of superheated fluid at high pressure that is deliberately placed in a metastable thermodynamic state. A particle that interacts with the medium and deposits energy above a threshold within a critical radius will result in the formation of an expanding bubble in the medium. An interaction below threshold or one that is more diffuse in the medium, spread beyond the critical radius, will form bubbles that immediately collapse. These chambers can be tuned so that nuclear recoils induce the expanding bubbles, but electron recoils do not. For more details and examples of experiments that use these techniques see Sec. 7.4.

The rich set of signals that result from the variety of direct detection experiments lends themselves to a variety of analysis approaches. These range from simple "box cuts" - a set of one-dimensional selection criteria on a set of variables $\vec{x}$ that each have sensitivity to ER, NR, or combinations of these effects - to multidimensional approaches like profile likelihood analysis and machine learning.

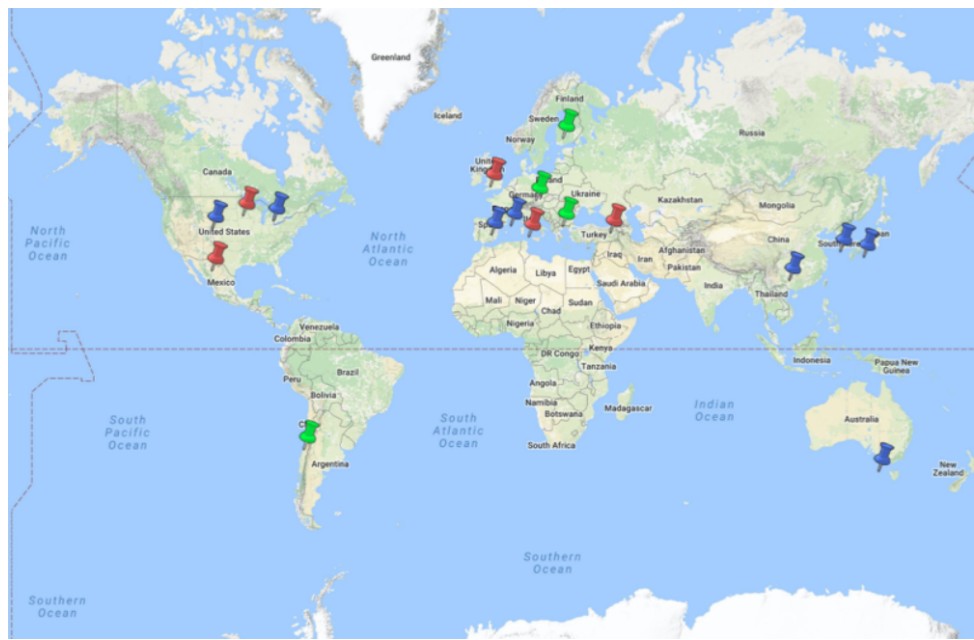

Figure 15: A map showing the 17 current underground facilities that can house dark matter experiments, as well as other scientific payloads that benefit from a location below the Earth's surface.

## 5 Detector Planning and Design

In this section, I will explain the many choices that have been made in the planning and design of dark matter direct detection experiments in response to the nature of the WIMP signal and the challenges posed by the many possible background processes.

### 5.1 Underground Laboratories

As noted earlier, the key means by which substantial cosmic ray backgrounds are mitigated is to site detectors, as well as construction facilities, deep underground. There are presently 17 sites world wide that are utilized as, or designed to be, underground laboratories (Fig. 15). The background intensities from muons and spallation neutrons are shown in Fig. 16 as a function of depth, with laboratory sites indicated as labels at their depths.

A key unit of depth measurement is the meters-water-equivalent (mwe). This unit is needed to fairly compare two different sites with two different overburdens. One lab may sit below a mountain of dense rock, while another laboratory may sit below an equivalent depth of loose rock and soil. These depths, stated directly, do not fairly compare the shielding differences of the two overburdens. Instead, geological surveys of the overburden are conducted and the densities and relative amounts of differnt materials is accounted. This information is then converted into the equivalent depth of a body of water, in meters, that would be represented by the combined shielding capabilities of the lab's overburden. This is the origin of the mwe unit.

The hadronic component of the cosmic ray flux is negligible after only $\mathcal{O}(10\text{ mwe})$ of overburden. Pions, kaons, and other light hadrons produced in nuclear interactions by cosmic rays are attenuated quickly by virtue of their electromagnetic and nuclear interactions. However, cosmic ray muons can penetrate far deeper and can produce high energy neutrons (*fast neutrons*). These, in turn, can enter the detector volume and result in keV nuclear recoils in detector atoms. Neutron production in the vicinity of the detector is facilitated by a few

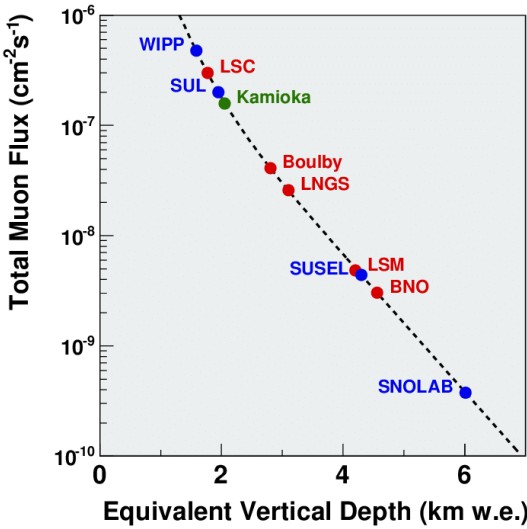

Figure 16: The intensity of muons vs. depth, with labels indicating the depth locations of various underground laboratory facilities. Taken from [24]

processes: (a) $\mu^-$ capture (an inverse beta decay reaction that can up-convert a proton to a neutron, which is then ejected from the nucleus); (b) photo-nuclear reactions initiated by electromagnetic showers in the surrounding laboratory materials; (c) deep-inelastic muon-nucleus interactions that "cleave off" neutrons from the nucleus; (d) hadronic interactions of nucleons with pions and kaons produced in hadronic showers induced by cosmic ray activities.

The challenge to the experimentalist is to anticipate fast neutron backgrounds, recognizing that surrounding overburden material as well as well-intentioned detector shielding materials (e.g. lead) can present production targets for fast neutrons. The experimentalist must then identify ways to either predict the rate of such backgrounds or veto them by passive and active means.

## 5.2 Activation of Detector Materials

Activation of a detector material itself or materials that end up close to the detector is another concern. This can happen during one or more of phases of detector design, fabrication, transport, assembly, and operation. For example, raw materials used in detector construction, even if originally purified to remove radioisotopes, can be activated by cosmic ray interactions during transport at the surface of the Earth. In addition, routes and locations on the surface can matter. For example, cosmic ray spectra vary with geomagnetic latitude and the flux varies with height above Earth. Choosing a mountainous shipping route, for example, may lead to much higher activation of materials than if the same material were transported along a route that is close to sea level.

Another challenge in the activation of materials is that there is, despite nearly a century of detailed work on radioisotopes, an incomplete understanding of the cross-sections of interactions that lead to production of isotopes. It is simply a fact that not all such production processes and their rates have been measured. In general, radioisotope production is dominated by $(n, x)$ reactions (95%) and $(p, x)$ reactions (5%).

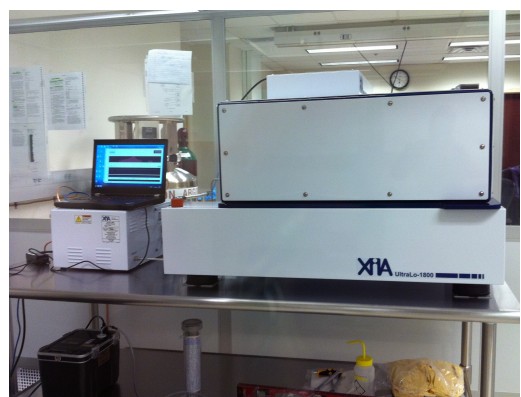

(a) The XIA Alpha Screening Facility at SMU. Photo provided by the SMU LUMINA laboratory.

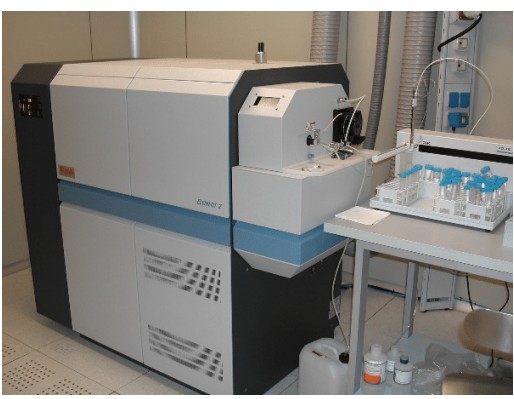

(b) ICMPS at the Gran Sasso Laboratory in Italy [25]

Figure 17: A small sample of the many facilities at universities and laboratories for screening materials before they are used in final construction of a dark matter experiment. Taken from [26].

## 5.3 Shielding from Environmental Backgrounds

A combination of high-atomic-number (*high-Z*) and low-*Z* materials are employed to diminish the neutron and gamma fluxes. Commonly employed materials are lead (prized for its density of target nuclei), polyethelyne (prized for its high neutron capture cross-section owing to the many hydrogen atoms in its molecular structure), and copper. In addition, purging volumes with gases like nitrogen can help to drive out airborne radioisotopes like radon gas.

Active and passive shielding is a key part of dark matter detector design. Some experiments employ large water shields to passively reduce or moderate both environmental radioactivity (e.g. gamma rays from surrounding unstable nuclei) and muon-induced neutrons (e.g. resulting from the presence of hydrogen in the water whose unpaired nuclear proton can readily capture a neutron). Current efforts have determined that a water shield with a thickness of 1-3 m can reduce underground fluxes of gamma radiation or radiogenic fluxes by a factor of $\sim 106$. An active muon shield, employing materials doped with scintillators (e.g. boron, gadolinium, etc.) that emit light when traversed by charged particles, can be used to identify background-triggering events. For example, a nuclear recoil induced by a neutron rather than a WIMP) could occur when a muon passes through the detector and spallates a neutron into the bulk detector material. An active muon shield would "tag" the passage of that muon with high efficiency, allowing experimenters to veto from consideration any nuclear recoil activity within a time window of the muon's passage.

The active dark matter detector volumes, nestled within a layer of active or passive shielding, can also provide the own shielding capabilities. This is known as *self-shielding*. For example, a large volume of liquid xenon employed for ionization and scintillation observations will generally absorb exterior backgrounds close to the surface of the xenon volume. Experimentalists who design such detectors with good three-dimensional position capabilities can identify these *surface events* and veto them. This is typically done by using external calibration sources to bombard the material, observing where most of those interactions then occur and defining a veto region in some outer volume of the active material. Of course, this self-shielding effect comes at a cost: the active mass of the detector is reduced by discarding a volume from consideration in the actual dark matter search. To compensate for the loss of target mass, experimentalists may have to run the experiment much longer (see Eqn. 74).

## 5.4 Internal Radioactivity in Detector Material

No amount of external shielding can protect a dark matter experiment from itself. For example, any uranium, thorium, potassium, cesium, cobalt, argon, or krypton (to name just a few) atoms embedded either in the active detector material or nearby shielding material can result in radioisotope decays that generate backgrounds. Most of these atoms are naturally unstable in their abundant isotopes, and some of them have unstable radioisotopes as a small fraction of their more abundant stable isotope(s).

Elimination of these radioisotopes from the construction process requires a diverse portfolio of techniques. In some cases, purification techniques can be used in situ, as is the case for experiments using liquid xenon as a target. Large distillation columns are used to purify the liquid xenon, bringing the levels of krypton and radon into acceptable ranges. When in situ purification is not feasible, materials have to be characterized and selected prior use in the construction. Care needs to be taken to avoid activation after construction, and detailed handling records (including altitude, duration of time spent outside of shielding, exposure to atmosphere, etc.) aid in later recognizing materials that may have become overly contaminated. This challenge is made particularly large because modern dark matter experiments are now so sensitive that radioisotope backgrounds have to be reduced below the 1 ppb level.

Active screening of materials (e.g. by placing materials in drift chambers or other detector instrumentation) is a part of this process. For example, highly sensitive alpha counters (Fig. 17a) can be used to estimate how much activity is implanted at or near the surface of small amounts of detector material. This activity of samples can then be extrapolated to the activity of all detector volumes constructed from this material. Another highly sensitive technique is inductively coupled plasma mass spectrometry (ICPMS) (Fig. 17b) which can detector very low levels of radioisotopes in samples. A combination of these approaches is the most common approach to insure compatibility between independent means of surveying material contamination.

Screening facilities utilize combinations of two major practices to establish a workflow for material assays. One approach is to augment commercially available systems. For example, a commercially purchased High-Purity Germanium (HPGe) detector can be placed in a custom-designed shield consisting of lead, copper, neutron moderation and capture materials, an active cosmic ray veto, and/or an underground location. Alternatively, a fully custom system can be developed by a research group or a multi-institution collaboration. Here, in addition to a custom shield, a custom cryostat design will be utilized with attention to design and placement of electronics to minimize background sources (such as uranium, thorium, or potassium). A comparison of these approaches is shown in Fig. 18. Many screening options exist (Fig. 19), and no doubt many more will be needed and developed as detector technology sensitivity continues to advance to meet the challenge of detecting dark matter interactions.

## 5.5 Modeling Radioisotope Backgrounds

Screening materials affords a chance to estimate radioisotope contamination of small samples of the detector. Material handling records and tracking affords a chance to estimate exposure to natural radioisotopes, like radon, or activation by cosmic rays. However, it is impossible to monitor the whole instrument in situ while it is operating. Inevitably, a computational model of the possible backgrounds and their impact on the experiment must be developed. These models employ fundamental physics, including the details of particle interactions in material (cross sections); data from material screening and material handling information; data from the experiment itself; and experience from past phases of experimentation with earlier instrumentation.

Three software frameworks exist to calculate the spectra of neutrons produced by $(\alpha - n)$

| Isotope/ Chain | Standard Size (ppb) \| (mBq/kg) | | Large Size & Long Count (ppb) |
|---|---|---|---|
| $^{238}$U | ~0.1 | ~1.0 | 0.009 |
| $^{232}$Th | ~0.3 | ~1.5 | 0.02 |
| $^{40}$K | ~700 | ~21 | 87 |
| $^{238}$U | 0.001 | 0.12 | |
| $^{232}$Th | 0.001 | 0.004 | |
| $^{40}$K | 1 | 0.031 | |

Figure 18: A comparison of material screening capabilities for various isotopes showing the difference between augmented commercial approaches (top three rows, light red) and custom approaches (bottom three rows, light yellow). Standard-sized samples are compared between both approaches, while large-sized samples with long counting times to integrate radiation exposure to the instrumentation is shown only for an augmented commercial approach.

interactions.

- SOURCES - (using the EMPIRE2.19 libraries for cross section inputs)

- USD WebTool (TENDL 2012 libraries which are validated by TALYS for cross section inputs)

- NeuCBOT (utilizing TALYS for cross section inputs)

TENDL is a validated library and EMPIRE is recommended by the International Atomic Energy Agency, but neither can properly calculate all resonant behavior that is experimentally observed. The output energy spectra from these simulations can be used in a full simulation of the whole experiment to predict the number of background events from neutrons. Validation of these approaches [27] is essential before applying them to background estimates in current or future experiments. For example, SOURCES-4A and USD agree on the neutron yield (neutrons/s/cm$^3$) from copper metal within about 20% of one another (including separate predictions of the neutrons from the $^{238}$U and $^{232}$Th decay chains). However, for a material like stainless steel these only agreed within about 50%.

Another example is to estimate the neutron fluxes from radioisotopes in photomultiplier tube (PMT) glass [28]. PMTs are a common technology for detecting and amplifying light emissions in detectors. The NeuCBOT (SOURCES-4C) framework would predict 15 (13) n/year from PMT glass. Within the ranges of constraints on these calculations and uncertainty on the underlying processes, these are fair comparisons.

Predicting neutron fluxes from materials is still a place where significant improvements are needed to keep up with the advances in detector sensitivity.

| Technique | Sensitivity |
|---|---|
| Radon Emanation | 0.1-10 Bq/kg (Ra) |
| Immersion Whole Body Counters | $10^{-13}$-$10^{-14}$ g/g (U/Th) |
| ICPMS<br>(Inductively Coupled Plasma Mass Spectrometry) | ppt to ppt (U/Th/K) |
| SIMS/GDMS<br>(Secondary Ion & Glow Discharge Mass Spectroscopy) | 1 ppb (SIMS)<br>10-100ppt (GDMS) |
| AMS<br>(Accelerator Mass Sepctroscopy) | < 1 ppt |
| Neutron Activation Analysis | 100 pg (U), 10 ng (K) |

Figure 19: A comparison of material screening approaches and their current standard sensitivities.

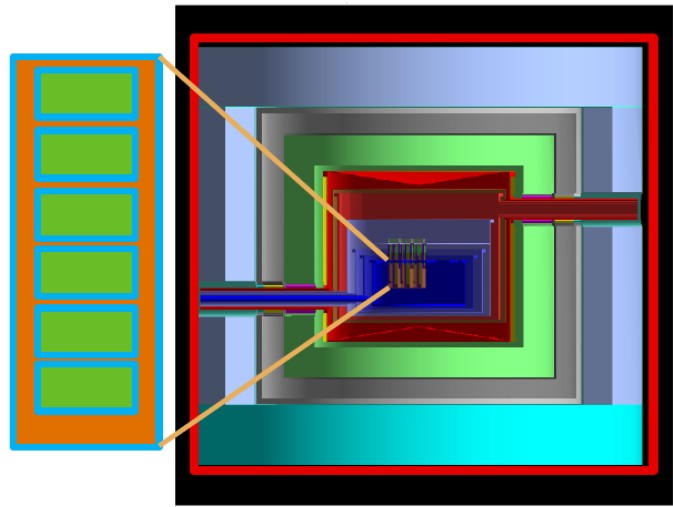

Figure 20: A GEANT4 model of the SuperCDMS dark matter detector. Image provided by the SuperCDMS Collaboration.

Simulation frameworks like GEANT4 can be used to develop detector and material geometry and response models. Those models can then be subjected to the expected fluxes of radioisotope backgrounds. An example of such a model for the SuperCDMS dark matter search experiment is shown in Fig. 20. SuperCDMS is a good example of the layouts I have been describing so far in this text. An exterior series of shields that include lead, polyethylene, and ancient lead depleted of radioisotopes encloses the cryostat (shown in red) which houses the active dark matter detector volume and maintains it at micro-Kelvin temperatures. The active detector volumes are stacks ("towers") of hockey-puck sized single crystals of Germanium. Each crystal is photolithographically etched with a pattern of sensors for ionization and phonon readout.

All of this complexity can be laid out in GEANT4 including gaps, services, and other realities of the detector. Each material can be modeled, along with its expected or measured level of radioactivity. A simulation of what the detector would see over a period of time can then be generated. From that the energy recoil spectrum purely from background processes can be produced. This can include radiogenic backgrounds, cosmic ray-induced backgrounds, and

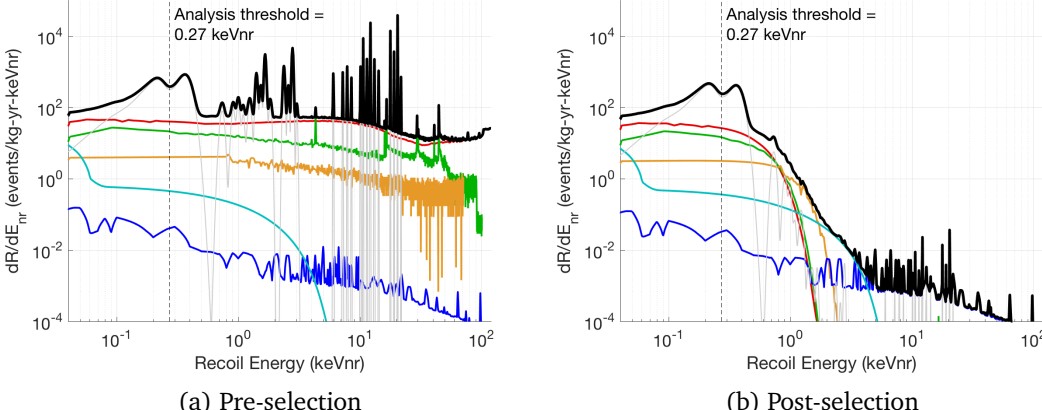

(a) Pre-selection                    (b) Post-selection

Figure 21: Figures from Ref. [29]. Background spectra, before (left) and after (right) analysis cuts in Germanium iZIP detectors, shown as a function of nuclear recoil energy (keVnr) to allow direct comparison of the various backgrounds to the natural dark matter interaction energy scale. Thick black lines represent the total background rates. Electron recoils from Compton gamma-rays, $^3$H and $^{32}$Si are grouped together (red). The Ge activation lines (grey) are shown convolved with a 10 eV r.m.s. resolution. The remaining components are surface betas (green), surface $^{206}$Pb recoils (orange), neutrons (blue) and neutrino interactions (cyan).

neutrino backgrounds and others not yet discussed here (Fig. 21). Signal events (dark matter nuclear recoils) will be defined by experiment- and analysis-specific thresholds; in the example shown, a dark matter nuclear recoil is counted only for $E_R > 0.27$keVnr. A 10 GeV/c$^2$ WIMP candidate can deposit up to $\mathcal{O}(10)$ keVnr; for the highest such recoil energy regions these models predict just fractions of a background event per kilogram of material, per keVnr, per year.

## 5.6 Designing Radiopure Materials

It is increasingly the case that experiments cannot use commercial-grade materials in the construction of experiments. For example, let's consider copper. This metal is essential in cryostat design and in the construction of shielding for sensitive electronics. Mined from the Earth and refined, copper is naturally contaminated by radioisotopes, primarily uranium and thorium. In addition, radioactive elements such as radon are known to implant on copper when it is left in open air. Radiopure copper is an essential ingredient in dark matter detector construction, but sourcing this material has grown more difficult as the sensitivity of dark matter detectors have been forced to increase.

The philosophy of physicists in this kind of situation is straight-forward: *if you cannot buy it, make it yourself.* For example, a facility at the Pacific Northwest National Laboratory (PNNL) has been constructed to electroform copper in a controlled manner. This controlled process can prevent naturally occurring contaminants from playing a significant role during the formation of the metal from a bath of copper ions (Fig 22). This manufacturing process results in copper with $\leq 0.1$ $\mu$Bq/kg activity levels from either the uranium or thorium chain.

Another example is the purification of noble gases and liquids for use in detector volumes. Xenon is utilized both in dark matter and rare neutrino search experiments. High purities are necessary in order to suppress the radioisotope backgrounds that would otherwise render unusable these instruments. The XENON Collaboration has designed, built, and operated their own xenon purification facility [30]. This system can both purify and distill the noble

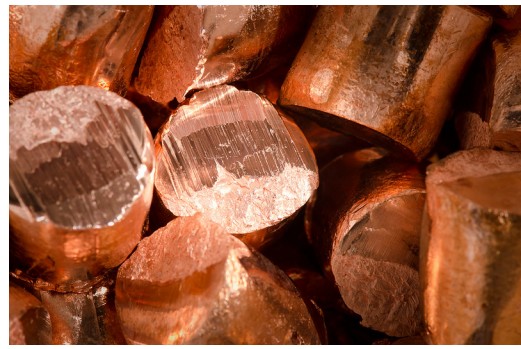

(a) Copper nuggets prior to electroformation

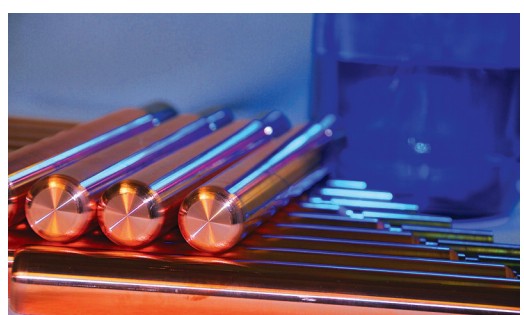

(b) Copper cylinders from electroformation

Figure 22: (Left) Copper nuggets prior to their use in copper electroforming. (Right) Copper cylinders resulting from the electroformation process. The blue solution in the background is copper sulfate from the electroforming bath in which these copper pieces were grown. Image provided by PNNL researchers Eric Hoppe, Brian LaFerriere, Jason Merriman and Nicole Overman and made available courtesy of Pacific Northwest National Laboratory.

gas. Whereas commercial xenon is contaminated by krypton at the level of 1 ppm - 10 ppb, the XENON 1-ton detector depends on levels at or below 0.2 ppt to meet its design goals. This is approximately 100,000 - 500,000 times higher purity than commercial-grade xenon. This is only achievable with their custom distillation and refinement approach, consisting of a 5.5 m distillation column that can generate 6.5 kg of refined xenon each hour.

## 5.7 Neutrino Interactions in Dark Matter Detectors

The neutrino is a low-mass, weakly interacting particle. In principle, it's a perfect test of whether or not a dark matter detector is capable of observing a weak-scale nuclear interaction like those we hope occur between dark matter and standard model matter. The challenge with neutrinos is that they are relativistic – they travel at nearly the speed of light – because of their extremely low masses. The sum of all three neutrino mass-eigenstates is constrained by the CMB to be less than 0.12 eV [31]. While the individual mass eigenstates' masses are not known, clearly they are a low-mass WIMP-like candidate at best.

The most copious nearby source of neutrinos is the Sun. The fusion reactions in its core, particularly the $pp$ chain of reactions, generate high fluxes of low-energy neutrinos (Fig. 23). These primarily contribute to the ER background via $\nu-e$ scattering at a level of 10 - 25 events per tonne of detector material per year at low energies. The lower-rate $^8$B fusion neutrinos have higher energies, and as a result their NR can not be distinguished from WIMP signals. Those are expected to lead to 1,000 events per tonne of detector material per year of operation. This will especially affect experiments using heavy targets such as germanium.

Finally, there are atmospheric neutrinos produced by cosmic ray interactions (e.g. muon decay in the atmosphere). These collisions produce pions which subsequently decay to muon and electron neutrinos and antineutrinos. Current and upcoming direct detection dark matter experiments are primarily susceptible to the component of the atmospheric neutrino spectrum whose energy is less than 100 MeV. There is also a diffuse background of neutrinos from supernova explosions. There also exists higher energy neutrinos than even the $^8$B but their flux is even lower. These are expected to contribute at the level of 0.01 - 0.05 events per tonne per year [32].

As direct detection technologies continue to advance, the neutrinos will provide a "fog" of electron and nuclear recoils that will have to be modeled with increasing precision. It will

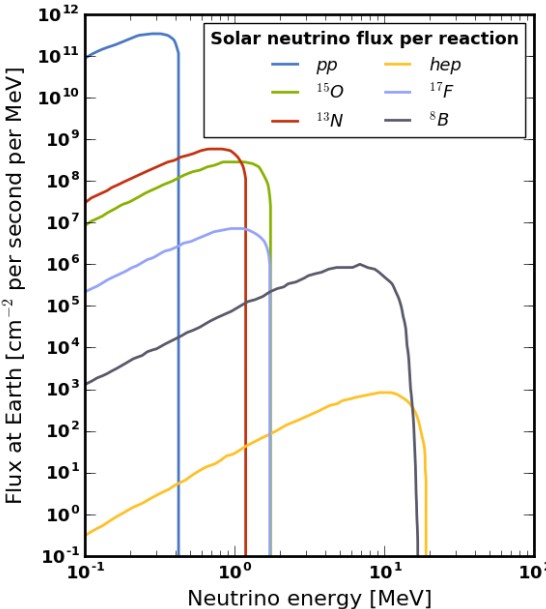

Figure 23: A few components of the solar neutrino flux spectrum.

need to be measured using dedicated instrumentation or in situ, and subtracted from the data to isolate any WIMP nuclear recoil signatures. They are not, at the time of the writing of this text, currently a problem. However, they are clearly on the horizon.

Directional detection capability may prove to be a crucial tool in future experiments for rejecting the solar neutrino background. Neutrinos from the Sun will, ideally, result in directional interactions, e.g. an ejected electron or recoiling nucleus, that points back toward the Sun. However, the weak interactions that, for example, cause a neutrino to scatter an atomic electron will result in multiple final-state particles and doesn't guarantee perfect alignment of the electron with the original neutrino direction.

The degree of directional correlation has been explored at leading order in the weak interaction [33]. The correlation between the scattering angle of the final-state electron and the incident angle of the original neutrino, as a function of the energy of the neutrino and folding in the solar neutrino spectrum, is illustrated in Fig. 24. This plot is based on the cited work, but is not published and is provided via an internal communication. There is a reasonable expectation that, at the 68.3% (95.4%) confidence level, the scattered electron will be within 1 radian (1.4 radians) of the incident neutrino direction. This may be sufficient information for future directional detection experiments to veto or categorize such events as more neutrino-like than WIMP-like.

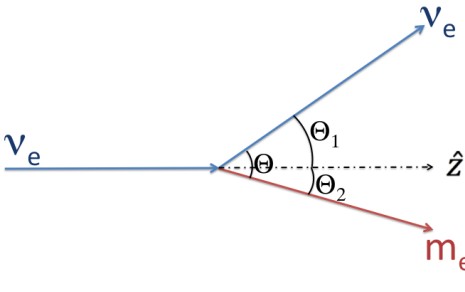
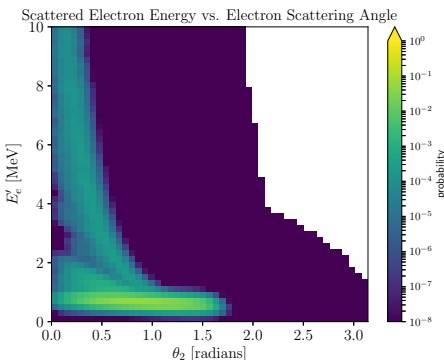

Figure 24: (Left) A model of the leading-order scattering of an atomic electron by an incident neutrino (e.g. a solar neutrino). Image from Ref. [33]. (Right) The degree of correlation between the ejected electron energy and the scattering angle with respect to the incident neutrino direction (private communication).

# 6 A Summary of Needs for Direct Detection Experiments

Direct detection experiments will clearly need the following characteristics to make progress:

- The ability to see low-energy WIMP induced recoils (> 10 keV). This will require

    - Radiogenic purity;
    - Low energy thresholds.

- Ability to distinguish nuclear recoils:

    - Differentiate between electronic recoils and nuclear recoils;
    - Differentiate between alphas and nuclear recoils.

- Radiogenic and cosmogenic backgrounds mitigation

    - Passive and/or Active shielding from these backgrounds;
    - Position reconstruction and fiducialization;
    - Characterization of these backgrounds.

- Long exposures with long term stability

    - This will be especially true to observe annual and diurnal modulation

The search space for dark matter is, in principle, vast. A variety of theoretical models provide compelling candidates for dark matter, and the space of their possible properties (e.g. cross-section vs. mass) is shown in Fig. 25. A non-exhaustive list of models include NMSSM [34], Asymmetric Dark Matter [35], MSSM [36] and CMSSM [37]. Existing experimental constraints can be overlaid on this space to demonstrate what has, and what has not, been excluded by the current portfolio of projects. I will conclude this chapter with a look across several ongoing or planned dark matter direct detection experiments.

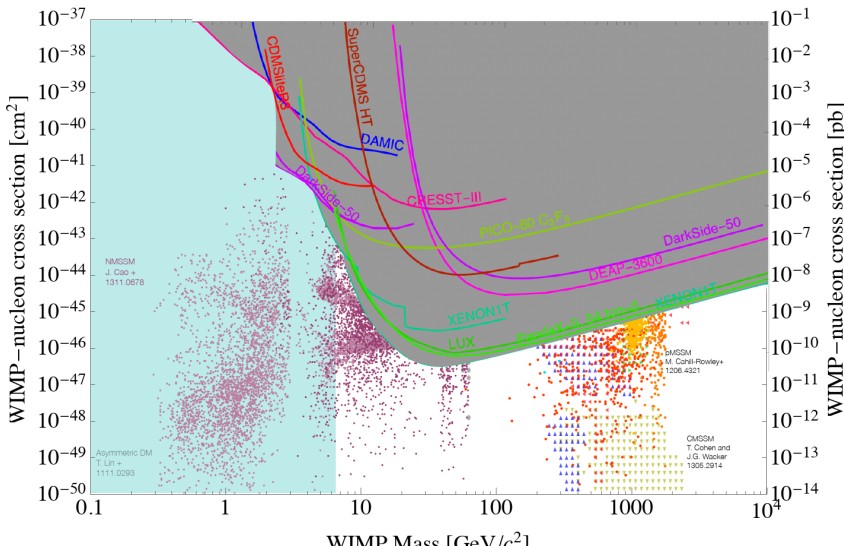

Figure 25: An example of the theory-driven search space for dark matter across a portfolio of plausible theoretical models for dark matter. Superimposed exclusions of that space based on current experimental efforts. Figure generated using [38].

# 7 A Sampling of Ongoing or Planned Direct Detection Experiments

## 7.1 Time-Dependent Annual Dark Matter Modulation: DAMA/LIBRA, COSINE, and ANAIS

A very fair question at this stage of the text is: *have we already seen strong evidence of a dark matter signature in any experiment?*. An illustration to the complexity of the subject of direct detection, is the unfolding series of results from the DAMA/LIBRA experiment and two independent instruments with sensitivity to the same proceses, COSINE-100 and ANAIS.

First DAMA, and then later DAMA/LIBRA, have been reporting positive results in the search for an annual modulated dark-matter-like signal since 1998. The DAMA phase of this experiment was a 100 kg NaI (Sodium Iodide) crystal array operated in Laboratori Nazionali del Gran Sasso from 1996 - 2002. The LIBRA phase of the experiment, operated by essentially the same collaboration of institutions, is a 250 kg CsI crystal array operating since 2003 with first results reported in 2008 and updated periodically since then.

The DAMA/LIBRA instrument is designed to measures scintillation from particle interactions in detectors. However, this signal is the only one from this instrumentation at the disposal of the experimentalists. This prevents them from being able to discriminate between a wide range of background hypotheses and the possible dark matter annual modulation signature. The primary effort of their work has been to search for such a time-dependent effect on top of a large background (e.g. a modulation with the right frequency and phase, see Section 3.8). This means accounting for time dependence in the possible backgrounds through simulation. A non-exhaustive list of potential backgrounds includes a time dependence in the annual cosmic ray rate, or in radioisotope backgrounds due to freeze/thaw cycles around the mountain laboratory that trap/release radioisotopes throughout the year. The experiment has no direct discrimination between nuclear and electron recoils.

They have reported a signal observed over 14 cycles with a significance above the null (no modulation) hypothesis of $12.9\sigma$ in the $2-6$ keV energy range (Fig. 27a). The collaboration is also able to lower their energy threshold with the new LIBRA phase of the experiment and have reported from that more limited sample alone a $9.5\ \sigma$ signal for single-scatter events in

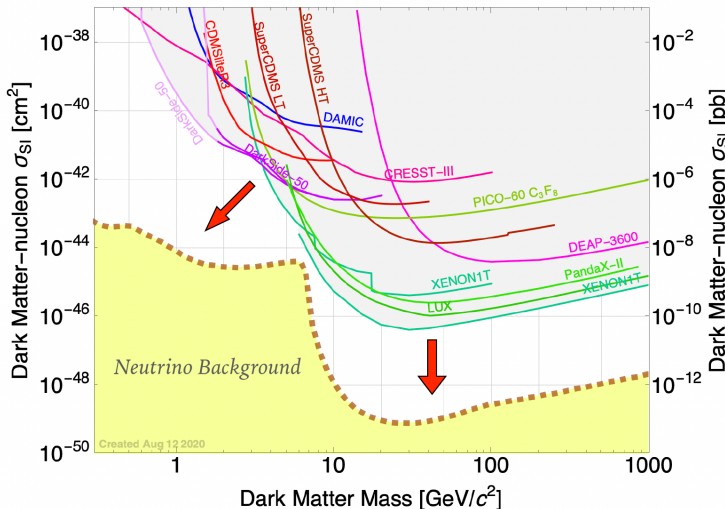

Figure 26: Experimental exclusion regions in spin-independent cross-section vs. mass space. The neutrino "fog" background expectation is shown in yellow at the bottom of the plot. The red arrows indicate directions that forthcoming experiments will push into the parameter space, toward better sensitivity at low mass and increased sensitivity at high mass. Image generated using [38].

the $1 - 6$ keV energy range observed over 6 cycles (Fig. 27b).

Again, the instrument has no in situ background discrimination capabilities. The collaboration has interpreted their observations in various ways, resulting in regions of the plane shown in Fig. 25 where we would expect to observe, with other independent instruments, those signals. However, independent experiments using very different technologies have not reported observing candidates with the expected masses and cross-sections. This has prompted a robust and extended debate over whether or not the DAMA/LIBRA background models are comprehensive, or whether or not the dark matter interpretation is tied to the NaI technology of DAMA/LIBRA and is not transferrable to other experiments.

Over the past two decades, with significant activity in the last five years, there have been efforts to construct experiments that can directly cross-check the claims of the DAMA/LIBRA collaboration. While earlier efforts may have been deflectable with arguments about dark matter models that allow for a signal in the DAMA/LIBRA experiment but no other running dark matter experiment, more recently independent collaborations have concentrated on developing high-purity NaI(Tl) crystals (to the level claimed by the DAMA/LIBRA collaboration in their own instrumentation) for use in similar scintillation-based experiments. Although I will focus on just two experiments with recent results whose methodologies are intentionally similar to DAMA/LIBRA, it should be noted that their are other collaborations also working on the cross-check including the SABRE project which plans to deploy detectors in both the northern and southern hemisphere.

The first is the COSINE-100 experiment [40] (Fig. 28). The instrument is located in Yangyang Laboratory in South Korea. It consists of 8 copper-encapsulated NaI(Tl) crystals. The total active detector mass is 106 kg. The detection of scintillation light is accomplished by two 3-inch PMTs per crystal, with an event trigger set at the level of a 0.2 photoelectron threshold. Calibration of the instrument is accomplished through insertion of sources close to the detectors via access tubes. The total background expected in this instrument is at the level of $2 - 4$ times the DAMA/LIBRA average, or aboe 2.7 counts/day/kg/keV on average in the same $2 - 6$ keV energy region. One favorable comparison to DAMA/LIBRA is that the in situ U/Th/K contamination in these NaI(Tl) crystals is below DAMA, while the $^{210}$Pb contamina-

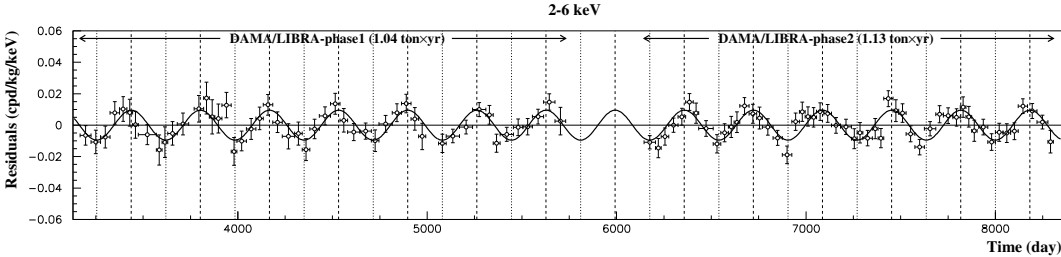

(a) DAMA/LIBRA Annual Modulation

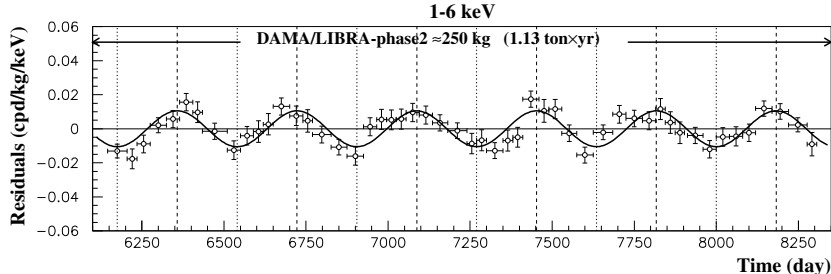

(b) LIBRA-only Annual Modulation

Figure 27: Results from the DAMA/LIBRA Experiment. (Top) The full 14-cycle DAMA/LIBRA annual modulation result, after subtracting flat background contributions. This analysis uses a higher energy threshold in the range of $2-6$ keV of recoil energy. (Bottom) The LIBRA-only results allowing for a lower-energy threshold of $1-6$ keV in recoil energy. Figures extracted from [39].

tion is at nearly the same level. This crystal detector design permits for a very high light yield during interactions in the material.

COSINE-100 has performed a similar annual modulation search. The signal is so obvious in the DAMA/LIBRA data that even with a modest exposure (e.g. just a couple of years) one should already expect to see this striking signal. COSINE-100 have published results from just 1.7 years (97.7 kg·years) of exposure. They perform a global maximum likelihood fit using a cosmogenic background and simple harmonic components fit simultaneously across all of the crystals, treating each crystal as an independent detector subsystem. The collaboration have excluded three of the crystals (crystal 1, 5, and 8) in this published analysis due to low light yield and excessive PMT noise. In regions of the data poor in anticipated WIMP signals ("sideband regions"), they observe an event yield in energy that decreases exponentially. This is consistent with known cosmogenic components.

The COSINE-100 Collaboration reported a best-fit result to their data that prefers an amplitude of $0.0092 \pm 0.0067$ cpd/kg/keV and a period of $127.2 \pm 45.9$ days, which is consistent with both the null hypothesis (no modulation) and with the DAMA/LIBRA best-fit value. This is still very early data from this specific experiment, and the collaboration projects that within 5 years of the start of data-taking they should be able to cover the DAMA/LIBRA signal region with $3\sigma$ significance from the null hypothesis. They have projected that future versions of their data analysis will lower the recoil energy threshold to 1 keV and improve on the selection of recoil events to reduce to total exposure time required to achieve the $3\sigma$ significance threshold for a DAMA/LIBRA-like signature.

The second experiment I wish to discuss is the ANAIS instrument [41]. This is also a NaI(Tl) crystal-based experiment, consisting of 112.5 kg of active material. The experiment

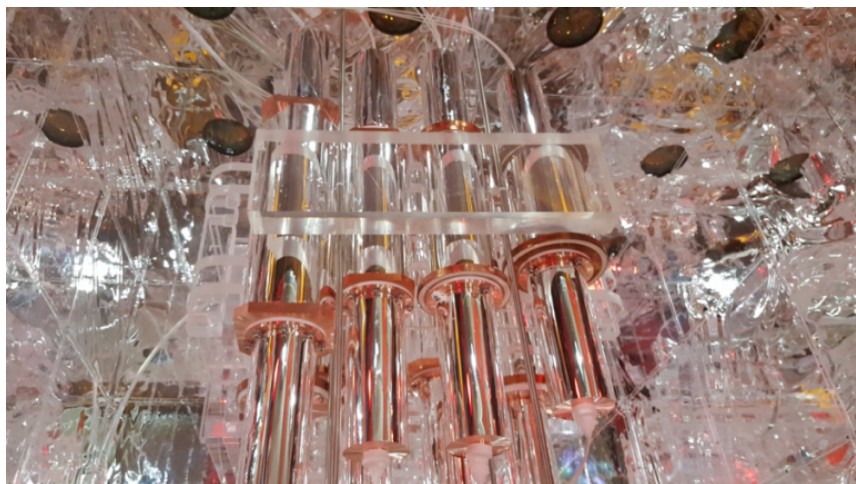

Figure 28: The COSINE-100 detector. Photograph courtesy of the COSINE-100 Collaboration.

is housed in the Canfranc Underground Laboratory (LSC), in Spain, and has been operating since 2017. In March 2021 they made public the results of an analysis of three years of data-taking (313.95 kg·y). They found a best fit in the [1-6] keV ([2-6] keV) energy region a modulation amplitude of ˘0.0034 ± 0.0042 cpd/kg/keV (0.0003 ± 0.0037 cpd/kg/keV), and determine that this is consistent with the absence of modulation signals in their data. Their results are compatible with the COSINE-100 and incompatible with DAMA/LIBRA result at $3.3\sigma$ ($2.6\sigma$) for an expected sensitivity to the DAMA/LIBRA-like modulation signature of of $2.5\sigma$ ($2.7\sigma$) (Fig. 29). While not completely definitive, taken together with the COSINE-100 results there is a growing body of evidence, using scintillation-based crystal experiments, that the DAMA/LIBRA result cannot be replicated as a dark matter signal.

## 7.2 Liquid Noble Detectors

Generally speaking, liquid noble detector technologies use a dual-phase (e.g. liquid and gas) approach to broaden the range of signatures at their disposal. This adds background discrimination capabilities, especially for high-mass dark matter candidates, that are fairly unmatched in the field right now. A few experiments that use this approach are XENON, LZ, Darkside, and PandaX. They operate the dual-phase systems as a time projection chamber (TPC), wherein electric fields are used to drift ions that result from nuclear and electronic recoils. A grid of electrodes provides two-dimensional information about the position of the the interaction when ions reach the electrodes. The third dimension is provided from the known drift speed of ions in the instrument, combined with information about when the original primary and secondary interactions occurred.

The principle of these instruments is straight-forward and based on significant experience with TPCs. Figure 30 illustrates the operations principle of this type of detector [42]). Interactions in the liquid phase (the largest single volume in the instrument) produce both atomic excitations and ionization as illustrated in Fig31. Excitation leads to an immediate pulse of scintillation light, establishing the $t_0$ of the interaction. This first signal is known as $S1$. Ionization electrons are drifted with an applied electric field and guided from the liquid phase into the gas phase. In the gas phase, electrons are further accelerated through a stronger electric field producing proportional scintillation. The pulse of energy from this effect is known as $S2$. PMTs on the bottom and top of the chamber record scintillation light. The pulse distribution in the PMTs provides the $x - y$ coordinates of the original interaction, while the drift time of

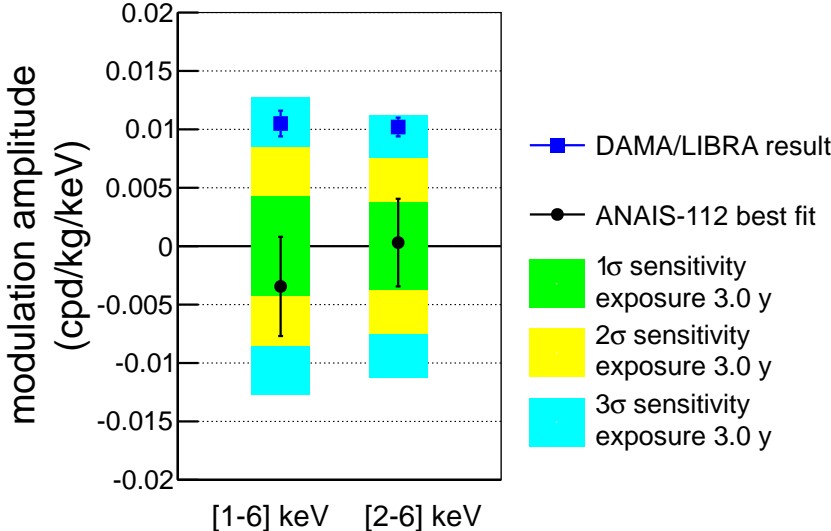

Figure 29: The ANAIS best-fit modulation amplitude result compared with the DAMA/LIBRA best-fit result for both recoil energy ranges considered by DAMA/LIBRA. Figure from Ref. [41].

the ions through the phases of the experiment yields the $z$ coordinate. The ratio of the energy deposited in $S1$ and $S2$ is a powerful discriminant against background processes, separating especially electron and nuclear recoils as a population.

For example, a gamma ray that ionizes an atom inside the liquid noble volume will result in a small pulse of initial energy (S1), and a comparatively larger pulse of secondary energy (S2) as the avalanche of ionization occurs when the original ionization electrons drift into the gas phase. A WIMP striking a nucleus in the liquid phase, however, will cause a large pulse of S1 energy followed by an equally large pulse of energy in S2. For the WIMP, $S2/S1$ will be closer to 1.0 than for the gamma ray.

The liquid noble experiments have come to dominate the search sensitivity for WIMP candidates with large masses $m_\chi \gtrsim 10$ GeV/$c^2$. This is evident from Fig. 26 where experiments like XENON1T, LUX, and PandaX define the parameter space exclusions in that mass region.

## 7.3 Cryogenic Solid-State Detectors

Cryogenic solid state detectors are crystal detectors are generally designed to readout both ionization and phonon signals from interactions withing the detectors. Compared to ionization detectors, phonon detectors can provide better energy resolution and smaller energy deposits in a variety of materials because many more phonons are generated in an interaction than electron-hole pairs. As an example, in semiconducting detectors an energy of $\sim 3-5$ eV is required to create an electron-hole pair. This equates to approximately 300 pairs per keV. As a comparison, scintillation detectors produce 20-40 eV per scintillation photon. When combined with the quantum efficiency of a photodetector is equates to to a few photoelectrons per keV.

There are two families of sensors for reading out phonon signals: thermal and athermal sensors. Thermal sensors measure an increase in temperature. A thermal sensor must wait for the full thermalization of the phonons within the bulk of the detector and the sensor itself, a process that occurs on the timescale of ms. Athermal sensors, on the other hand, measure fast, non-equilibrium phonons which contain information on the location and type of recoil that occurred. Schematically, the detector is comprised of an absorber with a weak thermal

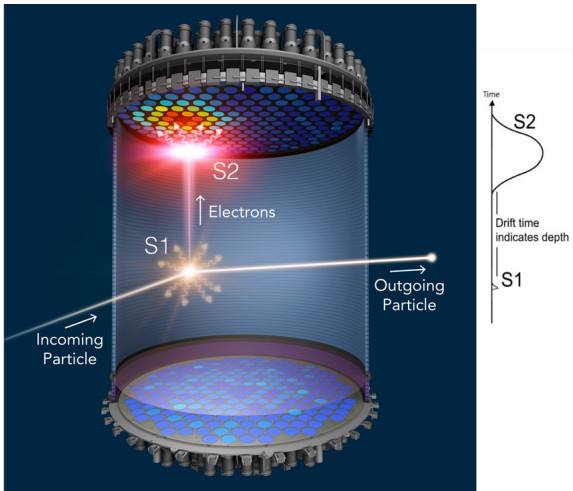

Figure 30: The operating principle of a liquid Xenon TPC using the XENON-1T detector as an example. Figure from Ref. [42].

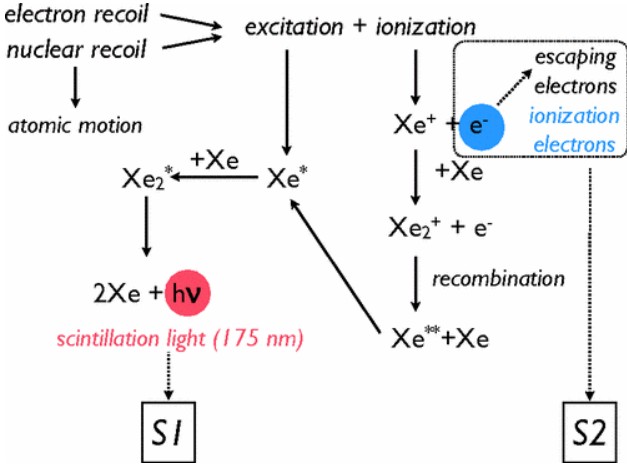

Figure 31: Schematic of scintillation production in two phase xenon detector. Ref. [43]

link to a refrigerator as illustrated in Fig. 32. The measured temperature increase in the cryogenic detector is given by

$$\Delta T = \frac{E}{C(T)} e^{-\frac{t}{\tau}} , \tag{115}$$

where E is the deposited energy, and C(T) is the temperature dependent heat capacity of the absorber. For pure dielectric crystals and superconductors at temperatures lower than their critical temperatures, the heat capacity is given by

$$C(T) \sim \frac{n}{M} \left( \frac{T}{\Theta_D} \right)^3 , \tag{116}$$

where m is the absorber mass, M is the molecular weight of the absorber and $\Theta_D$ is the Debye temperature. From this we see that the lower the crystal's temperature, the larger the $\Delta T$ per unit of absorbed energy. For an 100 g detector at 10 mK, a 1 keV energy deposition increases the temperature by $\sim$1 $\mu$K, a temperature increase that can be measured!

The two most widely used technologies to measure these signals are semiconductor thermistors such as neutron doped germanium sensors (NTDs) where the resistance is a strong

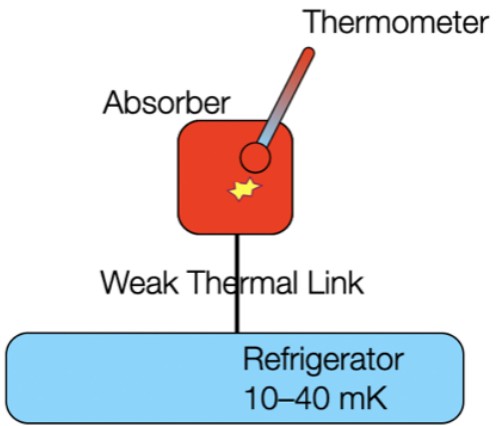

Figure 32: Schematic of a generic cryogenic detector. A thermometer measures the temperature of an absorber that has a weak thermal link to a refrigerator. The measured temperature increase is proportional to the deposited energy divided by the heat capacity of the absorber. Figure courtesy of Matt Pyle.

function of temperature and transition edge sensors (TESs). In both NTDs and TESs, an energy deposition produces a change in the electrical resistance that can be measured.

NTDs are small Ge semiconductor crystals that have been exposed to a neutron flux to make a large, controlled density of impurity. The resistance in these devices is strong function of temperature. NTDs measure small temperature variations relative to a $T_0$, which is set to be on the transition from the superconducting and resistance regime. The resistance is continuously measured by flowing current through and measuring the resulting voltage. The sensors are glued to the detectors.

A TES is a thin superconducting film that is operated near its critical temperature. TESs are fabricated onto crystal substrates using photolithography. A heater with an electrothemal feedback system maintains the temperature of the TES at its superconducting edge. When a particle interacts in the absorber (crystal target), phonons are generated and propogate to the surface of the absorber where they are absorbed by aluminum collection fins. When phonons enter the aluminum fins, Cooper pairs within the aluminum fins are broken, creating quasiparticles. Those quasiparticles diffuse into the TES, releasing their binding energy and increasing the temperature of the TES. This increase in temperature increases the resistance of the voltage-biased TES. Temperature changes are then detected by a change in the feedback current and collected by a Superconducting Quantum Interference Device (SQUID). This process is illustrated in Fig. 33.

### 7.3.1 SuperCDMS

The SuperCDMS collaboration has a long history of using TES technology for detector readout. Currently, the collaboration is constructing an upgraded Generation 2 (G2) experiment within SNOLAB in Sudbury, Canada. The cryostat will be able to accommodate up to seven towers and operate at a temperature of 15 mK. The initial payload will contain 24 detectors amounting to a mass of 30 kg. The detectors configured into four towers with each tower containing six detectors. Each detector will be a 100 mm diameter, 33.3 mm thick germanium or silicon crystal with mass 1.39 kg (Ge) or 0.61 kg (Si). Two of the detector towers will contain detectors operated in high voltage (HV) mode. Each HV tower will contain four germanium and 2 silicon detectors. The detectors in the remaining towers will be operated in interdigitated,

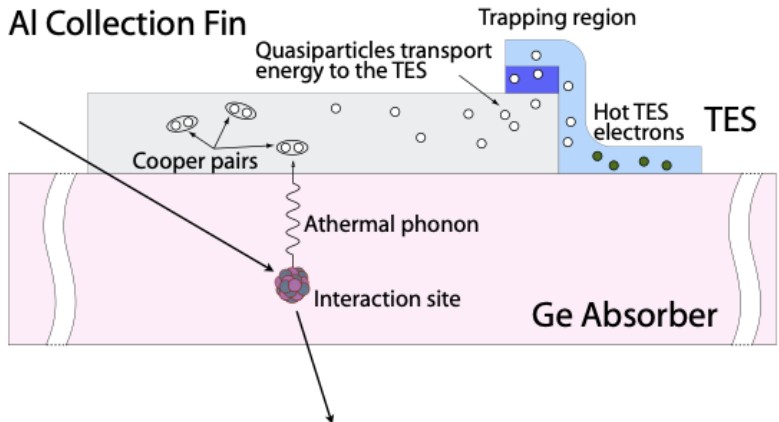

Figure 33: Schematic of a SuperCDMS phonon sensor on the surface of a Ge substrate. Phonons produced by particle interactions in the substrate break Cooper pairs in the aluminum fin producing quasiparticles. The quasiparticles diffuse into the TES increasing its temperature. Image provided by the SuperCDMS Collaboration.

z-sensitive ionization and phonon mediated (iZIP) mode. One of those towers will contain six germanium detectors and other tower will contain four germanium and 2 silicon detectors. The background from gamma particles is expected to be less than 0.1 dru (differential rate unit = event/kg/day/keV).

Each iZIP detector will have eight phonon channels and two charge channels on each detector face. The ionization sensors are interleaved between the phonon sensors on each face, as illustrated in Fig. 34. When a particle interacts in these detectors, simultaneous measurements of prompt phonon and ionizations signals allow for the robust discrimination between nuclear recoiling and electron recoiling events. One issue that was a concern in previous detector designs was that beta events occurring near the surface of the detector could mimic a WIMP interaction. The new interdigitated design allows for the identification of surface events by the distribution of the charge signal for a given event. If the event has symmetric charge distribution between the top face and the bottom face, it is a bulk event. If the charge is distributed primarily on one side of the detector, it is a surface event. As such, all surface and electron recoil events above a few keV can be removed using appropriate selection criteria. These detectors have sensitivity in a "background-free" mode to WIMPs with masses above $\sim$5 GeV and are sensitive to WIMPs with masses above $\sim$1 GeV in a "limited-discrimination" mode.

The SuperCDMS HV detectors will have only six phonon channels on each face to take advantage of the Neganov-Trofimov-Luke (NTL) effect. When electrons are drifted across a potential (V) a large number of phonons are generated. The total phonon energy ($E_t$) is a combination of the primary recoil energy ($E_r$) and the NLT phonon energy. The NTL phonon energy is given by

$$E_{NTL} = N_{eh}eV_b, \tag{117}$$

where $V_b$ is the bias voltage across the detector, the number of electron-hole pairs generated is $N_{eh} = E_r/\epsilon_\gamma$ and $\epsilon_\gamma$ is the average energy needed to generate an electron-hole pair. In germanium, ($\epsilon_\gamma$) is 3 eV. Thus the total phonon energy can be written as

$$E_t = E_r + N_{eh}eV_b. \tag{118}$$

When $V_b$ is very large, the NTL phonons dominate the signal and allow lower energy thresholds

SciPost Phys. Lect. Notes 55 (2022)

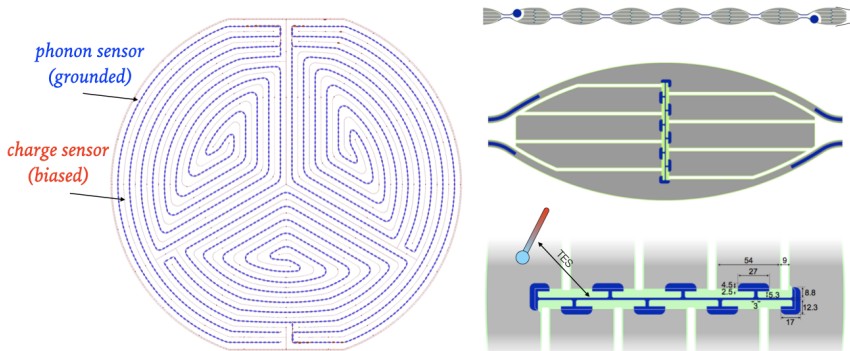

Figure 34: Schematic of a SuperCDMS iZIP detector. Left: Lines of ionization sensors (red) are interleaved between grounded phonon sensors (blue). Right: Each phonon sensor consists of an aluminum superconducting fin (grey) and a TES(blue.) Figure provided by the SuperCDMS Collaboration.

to be reached.

HV detectors provide ultra-high resolution by indirect charge measurements, but these detectors do not provide yield or detector face discrimination. The capabilities of HV detectors was demonstrated using the SuperCDMS Soudan CDMSlite detectors where thresholds of 75 eVee and 56 eVee were achieved as illustrated in Figure 36. The HV detectors to be operated in SNOLAB will be able to search for WIMP dark matter with masses down to ∼0.3 GeV.

As noted earlier, nuclear recoils produce electron-hole pairs less efficiently than electron recoils. As such the energy and interaction type can be defined through the yield (Y) where $Y \equiv 1$ for electron recoils.

$$N_{eh} = Y(E_r)\frac{E_r}{\epsilon_\gamma}. \tag{119}$$

Thus, the total recoil energy can be written as

$$E_t = E_r\left(1 + Y(E_r)\frac{eV_b}{\epsilon_\gamma}\right). \tag{120}$$

The energy scale is also dependent upon the interaction type. The detectors are calibrated using electron recoils and the resulting energy scale is labeled as keV$_{ee}$. To convert to the nuclear equivalent energy (keV$_{nr}$) by equating the equation above for nuclear recoils and electron recoil. Thus

$$E_{nr} = E_{ee}\left(\frac{1 + eV_b/\epsilon_{gamma}}{1 + Y(E_{nr})eV_b/\epsilon_{gamma}}\right), \tag{121}$$

where Y($E_{nr}$) must be determined by either direct measurement or with a model. The most common model used for ionization yield is the Lindhard model [45–47]:

$$Y(E_{nr}) = \frac{k \cdot g(\epsilon)}{1 + k \cdot g(\epsilon)}, \tag{122}$$

where g($\epsilon$) $= 3\epsilon^{0.15} + 0.7\epsilon^{0.6} + \epsilon$, $\epsilon = 11.5E_{nr}(\text{keV})Z^{7/3}$, $Z$ is the atomic number of the material and k $= 0.157$ in germanium. Although this value of k agrees with experimental measurements above 1 keV$_{nr}$, there are few measurements at lower energies. As such, for the lowest energy events an uncertainty must be taken into account. This is typically done by varying k uniformly between the majority of experimentally observed data [48–50].

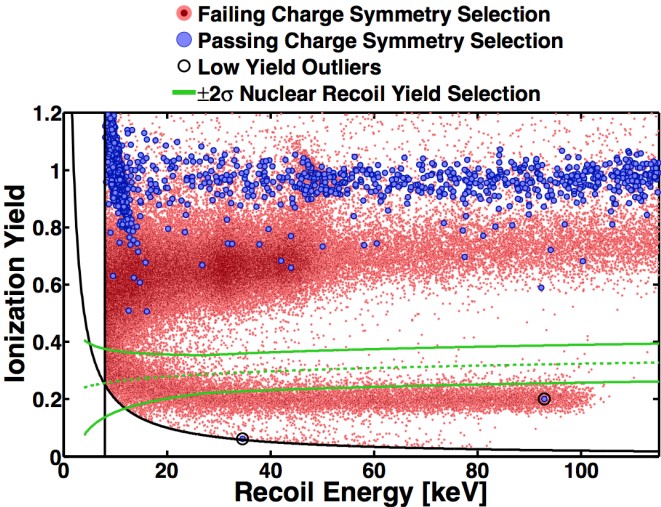

Figure 35: Ionization yield versus phonon recoil energy with the $\pm 2\sigma$ ionization yield range of neutrons indicated (area within green lines). Taken from 900 live hours of calibration data from with a $^{210}$Pb source facing one side of a detector. The hyperbolic black line is the ionization threshold (2 keVee); the vertical black line is the recoil energy threshold (8 keVr). Symmetric charge events (large blue dots) in the interior of the crystal and the events that fail the symmetric charge cut (small red dots) include surface events from betas, gammas and lead nuclei and are clearly separated from the nuclear recoil band. Figure provided by the SuperCDMS Collaboration.

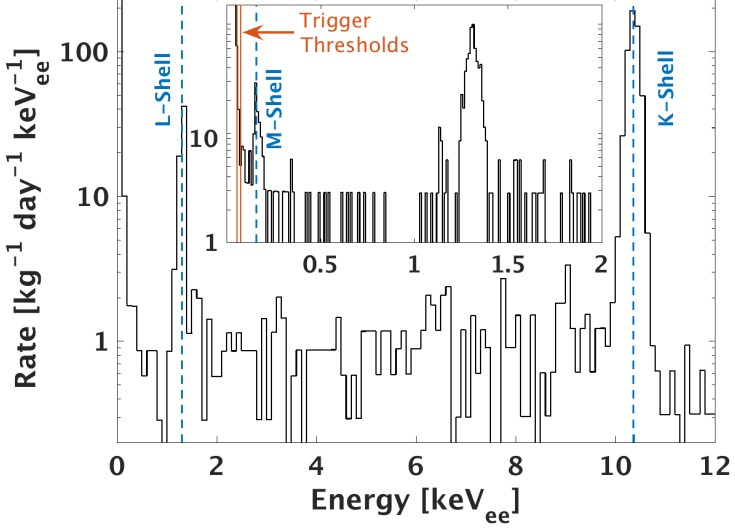

Figure 36: Illustration of the resolution of a SuperCDMS HV detector using the WIMP-search spectrum after application of all cuts and correcting for efficiency (except trigger efficiency) in the CDMSlite detectors. Inset shows a zoom (with smaller bin size) of the energy range actually used to the set the limit. $^{71}$Ge activation peaks are marked with blue dashed lines. Adapted from [44].

The SuperCDMS collaboration is also pursuing the development of HVeV detectors. These detectors have demonstrated single electron/hole pair resolution and have sensitivity to a variety of sub-GeV dark matter models using just gram-day exposures [60–62].

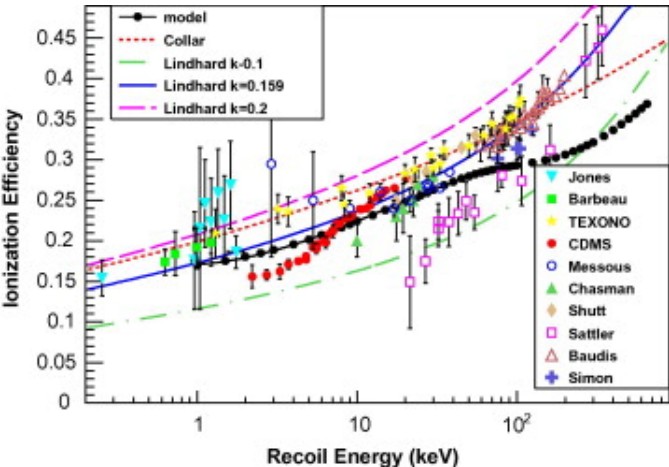

Figure 37: Comparison of ionization yield versus recoil energy for various models and experimental data. Models are Barker and Mei [48], Collar [50] and Lindhard [45] Data points are from Jones and Kraner [49,51], Barbeau et al. [50], TEXONO [52], CDMS [53] Messous et. al [54], Chaseman et al. [55], Shutt et. al [56], Sattler et al. [57], Baudis et. al [58], and Simon et.a l [59]. Figure adapted from [48].

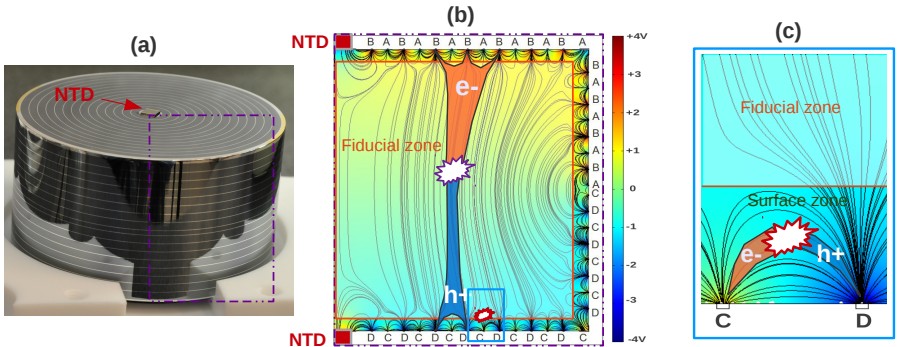

Figure 38: (a): An FID detector showing the interleaved ionization sensors and NTD sensor. (b): Cross-section of an FID detector illustrating the fiducial zone for an event generated in the bulk of the detector. (c): Illustration of charge collection for an event near the surface of the detector. Adapted from [63].

### 7.3.2 Edelweiss

The Edelweiss III collaboration has a long history of using bolometer-style readout technology. The experiment is located in Laboratoire Souterrain de Modane (LSM) and consists of 24 cylindrical high purity germanium crystals, each with mass ∼860 g operated at 18 mK [63]. Their fully interdigitated (FID) detectors collected charge via concentric aluminum ionization sensors that were interleaved on all absorber surfaces and two NTD Ge sensors that were glued onto the top and bottom face of the detectors. The configuration of the ionization sensors allowed for the fiducialization of bulk events in the inner region of the detectors as illustrated in Fig. 38. These detectors show strong separation on an event-by-event NR versus ER basis down to ∼5 keV as illustrated in Fig. 39. The rejection of ER events to NR events based on yield alone is $< 2 \times 10^{-5}$. The readout of all electrodes provides for a $< 4 \times 10^{-5}$ rejection of the especially concerning class of backgrounds, events occurring near the detector surfaces.

There are three issues that are somewhat limiting the Edelweiss experiment performance.

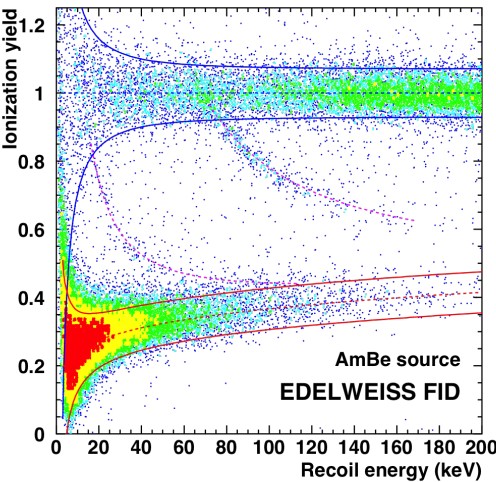

Figure 39: Ionization yield versus recoil energy for data taken with an AmBe neutron calibration source illustrating clear separation between electron recoils with yield ∼1 and nuclear recoils with yield ∼0.3. Blue lines indicate the 90% C.L. for electron recoils, red lines indicate the 90% C.L. for nuclear recoils, and the magenta lines illustrate the inelastic scattering of neutrons on the first (13.28 keV) and third (68.75 keV) excited state of $^{73}$Ge. Adapted from [64].

The most concerning for WIMP searches is that the neutron background is five times what is expected. It is too large for searches of WIMPs with masses >∼10 GeV/c$^2$. However, this neutron background is not of concern for searches of lighter mass WIMPs. Similar to Super-CDMS, the Edelweiss collaboration is developing a new detector called "RED" to pursue very light, electron interacting dark matter models. The NTD phonon resolution and an unknown background source of "heat-only" events (events that deposit energy in the detector without the creation of electron-hole pairs) are currently limiting factors in their searches [65]. These issues make it difficult to disentangle, noise-triggered events, possible events from leakage currents and other such backgrounds from potential dark matter signal candidates.

### 7.3.3 CRESST

The CRESST experiment is operated in the Laboratori Nazionali del Gran Sasso. The experiment uses an array of 10 cryogenic scintillating CaWO$_4$ crystals that are self-grown. Each crystal has dimensions of 20 mm x 20 mm x 10 mm and is instrumented with tungsten TESs to readout phonon signals. Each detector module consists of a scintillating crystal with mass of ∼25 g equipped with a TES next to a silicon on sapphire light absorber which is also equipped with a TES. The modules have fully scintillating housing and are instrumented "iStick" holders which provide the ability to veto radiogenic backgrounds from surrounding surfaces and suppress thermal signals from particle interactions in surrounding materials [66]. Similar to SuperCDMS and Edelweiss, the CRESST detectors allow for the discrimination between ER and NR. A schematic of the detectors is shown in Fig. 40.

The most recent results [67] from the CRESST detector were obtained with a single module. The combination of light sensors and the element oxygen in the detector target allows sensitivity to dark matter particle masses as low as 160 MeV/c$^2$. An AmBe neutron calibration source is used to define the NR band for each element in the crystal. Selection criteria was used to remove periods of abnormally high trigger rates due to electronic disturbances. Events that

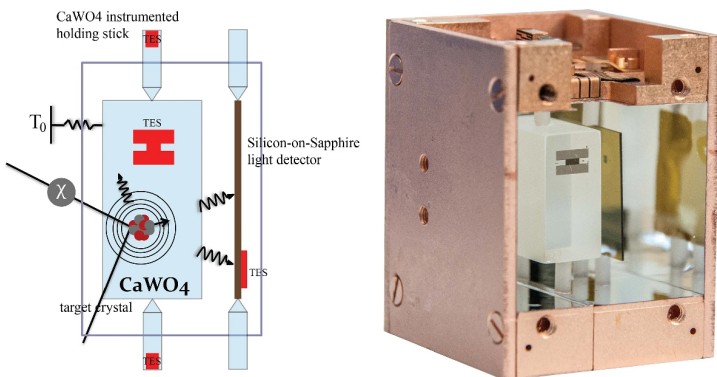

Figure 40: Left: Schematic of the CRESST-III detectors highlighting aspects of the improved detector design which includes fully scintillating housing and CaWO$_4$ instrumented holding sticks. Right: Picture of a CRESST-III detector module. Adapted from [66].

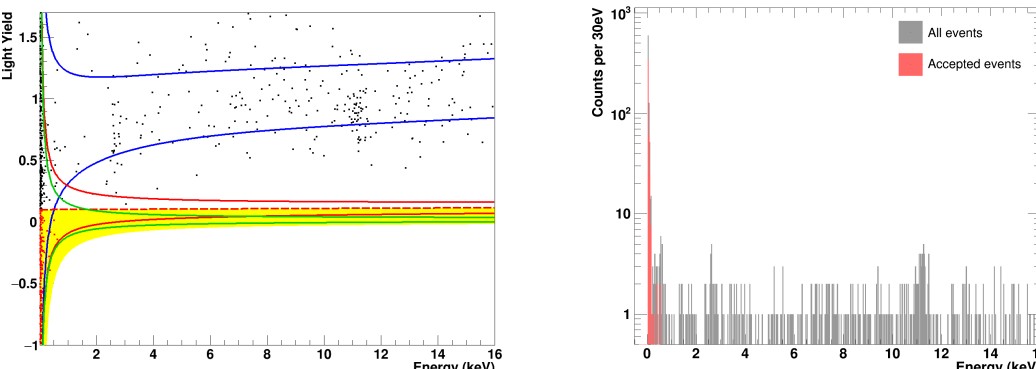

Figure 41: Left: Light yield vs phonon energy for events passing selection criteria in the dark matter data set. The blue lines enclose the 90% C.L. electron recoil band, the red and green lines enclose the 90% C.L. nuclear recoil bands for oxygen and tungsten, respectively. The yellow band encloses the dark matter candidate acceptance region. All events in this region are taken as dark matter candidates. Right: Energy spectrum of all events passing selection criteria in the dark matter data set. Candidate events in red fall within the dark matter candidate acceptance region. Adapted from [67]

triggered only the light channel were removed in addition to events of poor quality. Finally, events triggered in coincidence with the muon veto were removed. The livetime of the data set after all criteria were applied was 3.64 kg days. The energy and yield of the events passing all selection criteria are shown in Fig. 41.

All 441 events in the acceptance region were considered to be dark matter candidates with no background subtraction when setting the upper-limit on the WIMP-nucleon scattering cross section using the Yellin Optimum Interval method [68]. This result excluded new parameter space down to 0.16 GeV/c$^2$ and improved exclusion limits by a factor of 6 over the previous best limit.

## 7.4 Bubble Chambers and Superheated Liquids

Bubble chambers are sealed vessels filled with a liquid under pressure. To understand the principles of bubble nucleation [69], consider a bubble in thermal and chemical equilibrium.

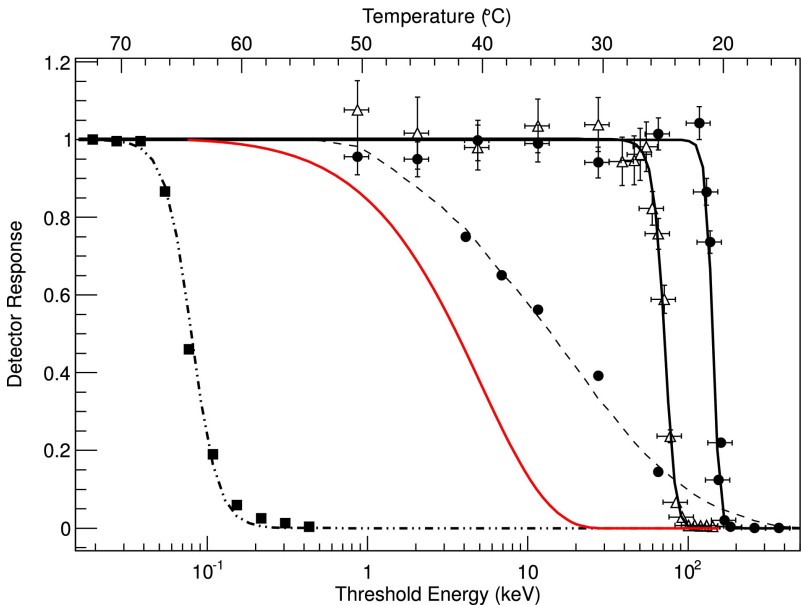

Figure 42: Detector response to different particles in superheated $C_4F_{10}$. From left to right: 1.75 MeV $\gamma$-rays and minimum ionizing particles (dot-dashed); model of 50 GeV/c$^2$ WIMP (red); polyenergetic neutrons from an AcBe source (dotted); $\alpha$ particles at the Bragg peak from $^{241}$Am decays (open triangles); and $^{210}$Pb recoil nuclei from $^{226}$Ra spikes(dots) Taken from [70]

.

In that case the temperature of the liquid is equal to the temperature of the bubble. Assuming that there is no surface tension, if the pressure of the bubble, $P_b$, is greater than the pressure of the liquid, $P_l$, the bubble will expand. If we include the pressure due to surface tension, $P_s$, the bubble will grow when

$$P_b > P_l + P_s \tag{123}$$

and the radius of the bubble is bigger than a critical radius. The pressure due to surface tension can be written as

$$P_s = \frac{2\sigma}{r}, \tag{124}$$

where $\sigma$ is the surface tension which is material dependent. Thus, we write the radius condition for bubble growth as

$$r > r_c = \frac{2\sigma}{P_b - P_l}. \tag{125}$$

Bubbles that do not meet these conditions collapse. The threshold for bubble nucleation is given by

$$E_T = 4\pi r_c^2\left(\sigma - T\left[\frac{d\sigma}{dT}\right]_\mu\right) + \frac{4\pi}{3}r_c^3\rho_b(h_b - h_l) - \frac{4\pi}{3}r_c^3(P_b - P_l), \tag{126}$$

where $\rho$ is the bubble density and h is the specific heat of the liquid ($h_l$) or bubble ($h_b$).

Bubble chambers as dark matter detectors are attractive because of their unique ability discriminate between different particle interactions [70]. Figure 42 shows the response of superheated $C_4F_{10}$ to various particles. Electron recoiling events have much lower thresholds compared to events that produce nuclear recoils. Thus, the detectors can be tuned to trigger only on nuclear recoiling events.

To mitigate nuclear recoils from neutrons, the experiments can be located underground and use active and/or passive shielding. Nuclear recoils from alpha events can be identified

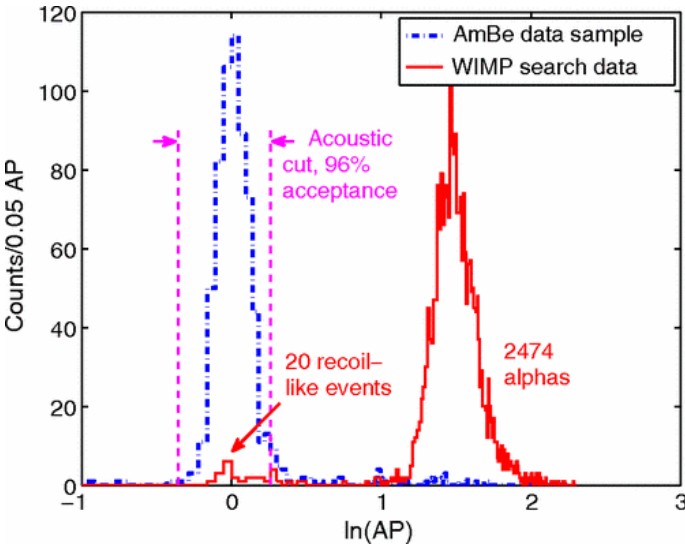

Figure 43: Detector response to alpha particles and nuclear recoils as characterized by an acoustic parameter (AP) in the COUPP-4kg dark matter experiment operated in SNOLAB. Adapted from [71].

using acoustic techniques. Alpha particles deposit their energy over tens of microns whereas nuclear recoils deposit their energy over tens of nanometers. The result is that alpha particles are ~4 times louder than neutrons and dark matter candidates, an effect that can be measured by piezoelectric sensors as illustrated in Fig 43.

The PICASSO and COUPP experimental collaborations have joined forces to create a new collaboration called PICO. The new collaboration has ambitions to ultimately build and operate a 500L bubble chamber. The collaboration has already produced results from two runs with PICO-2L [72,73] and PICO-60 [74]. Operation of these detectors pointed towards problems connected to the water piston-active fluid interface. As such, a redesign was implimented in the PICO-40L experiment where the superheated target sits on top of a fused silica piston.

# 8 Conclusion

The next decade will be very exciting for dark matter direct detection. The next stages of multiple large target experiments (called Generation 2 or "'G2" experiments) with sensitivity to the neutrino fog will be coming online. The XENON and LUX program two phased xenon experiments are expected to have sensitivity to or past the neutrino fog. Research and development for even more massive xenon and argon detectors are taking place within the DARWIN and GADMC collaborations. The solid state detector program is making fast progress in pushing their technologies to sensitivities to the lowest mass WIMPs.

Although WIMPs remain a viable and interesting candidate, other dark matter scenarios are gaining traction in the theoretical community and technological advances are allowing for the exploration of these ideas. This has opened up a new window of exploration for solid state cryogenic detectors.

Given the wealth of possibilities, a diverse set of experimental designs and targets are needed to constrain the theory and couplings of any discovered signal. As a community, we are well positioned to take on those challenges.

# Acknowledgements

I would like to thank the organizers of the Les Houches 2021 Summer School on dark matter physics for the opportunity to teach such an engaged bunch of students. I would also like to thank the students for their lively discussions and inquisitive questions which led to invaluable contributions to improving these notes. In particular I would like to thank Dan Salazar-Gallegos for his extensive editorial suggestions. I would also like to thank Stephen Sekula for discussions and editorial assistance.

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
