# Peer review of "Dark Matter Direct Detection of Classical WIMPs"

_SciPost Physics Lecture Notes, doi:SciPost Phys. Lect. Notes 55 (2022)_

## Round 1 · Referee Report · Anonymous (Referee 3) · 2021-11-25

Strengths
1) Level of the text appropriate for pedagocical lectures.
Weaknesses
1) The text discusses estensively physics associated to DM detectors while aspects as astrophysical inputs and theoretical interpretation of the signals are overlooked.
2) Figures.
Report
Dark Matter Detection is one of the most relevant topics in Modern Particle Physics hence, in this regards, the manuscript meets the acceptance requirements of the journal. However, only some aspects of DM detection are discussed extensively while others just recevied a small mention. Nevertheless, I think that this text would be a useful introduction, to some aspects of Dark Matter detection, for people new to the field, provided that some changes, are made to the text.
Requested changes
1) I would suggest a general proof-read of the text. There are some typos and some sentences which should be made more fluent to read.
2) Most of tables and figures are either too big or too small. In particular I would recommend, if possible, to replace fig. 13 with a plot with better resolution and to substantially increase the size of fig. 25 since, in its current status, the text in the panels could be read only by applying a substantiall zoom to the pdf reader.
3) Even if only a short section is devoted to the Dark Matter distribution I would recommend to include extra references besides just reference [1].
4) The acronym WIMP is used before being introduced. Furthermore at the beginning of the text the author that states the word WIMP will refer to just a generic particle candidate without specific assumptions on the mass scale and size of interactions. I do not agree with this choice; first I think it could lead potential confusion to someone interested in consulting the specialistic literature after reading these lecture notes. Furthermore, the lecture notes mostly focus on DM-nucleus scattering. Non-WIMP DM candidates can be probed also through different processes like scattering on electrons. More in general, in the second part of the text, when detection tecniques are discussed in more detail, the word WIMP seems to be used referring to its conventional meaning. For the reasons just illustrated I would recommend the author to modifiy appropriately the text and identify with better clarity which kind of particle DM candidates the discussion of the manuscript applies to.
5) The author correctly notices, after eq. (7), that the typical energy exchanged in DM scattering processes is not sufficient to break the nuclei. However, inelastic scattering processes in which nuclei pass from the ground to an excited state might occur, see e.g. Phys.Rev.D 96 (2017) 2, 022008.
6) I would reformulate the text around equation 88-89. It should be stressed that the $A^2$ enhancement occurs just for SI interactions. Furthermore, the relation between $f_p$ and $f_n$ depends also on how the DM interacts with up and down quarks individually so the assumption $f_p \sim f_n$ is model dependent.
7) Section 3.6 should be completed by showing how the differential rate shown in the previous sections can be expressed in terms of the effective operators.
8) The author should provide a more extensive description, in the main text, of fig. 25.

---

## Round 1 · Referee Report · Anonymous (Referee 2) · 2021-11-26

Strengths
1- Very comprehensive
2-Clearly written
3-Up-to-date
4-Self-contained
Weaknesses
1-The work is titled "Dark Matter Direct Detection of Classical WIMPs" but the discussion goes well beyond. This is in general the path that the community has taken (going beyond the Lee-Weinberg limit), but it is stated nowhere in the text.
Report
The lecture notes cover the very active field of direct searches of WIMPs under classical assumptions. It begins with theoretical background and discusses the complete framework under which experimental searches find their motivation. Then experimental approaches are discussed in details and the current scenario is very well presented. The discussion is complete and systematic and it certainly meets the criteria for publication in SciPost Physics Lecture Notes
Requested changes
1- Page 3 Only semiconductors are mentioned for detecting DM-nucleus scatting, but in the end of the notes other materials are discussed (i.e., CaWO4). In here the reader get the impression that semiconductors are the only viable option and this is not the case. Ic can be canced to Solid state detectors.
2- Page 9 typo "detector" -> "detect"

---

## Round 1 · Referee Report · Carla Macolino (Referee 1) · 2021-12-6

Strengths
1- it gives a complete and detailed overview of the WIMP dark matter direct detection current and future scenario
2-both the thoretical background and experimental aspects of this field are discussed extensively
3-it represents a useful and complete reference for students and scientists new to the field
4-the future experimental scenario for the direct detection of dark matter is discussed, putting emphasis on the challenges and capabilities of the different experiments
Weaknesses
1-some experimental aspects are not discussed, probably due to the vastness of the field
2-some minor experiments are not mentioned
Report
The review is a complete and detailed review on the WIMP dark matter direct detection scenario for current and future experiments. This manuscript appears to be very clear and sufficiently detailed to represent a valid reference for the knowledge of this field current status and future prospectives.
I definitely recommend the publication of this paper review, after a minor revision described below.
Requested changes
1-general: the Sun when referred to the name of the star in our Solar System should be written in a capital letter
2-page 10 the generates --> that generates
3-page 22: about-->around
3-page 23: interactions to observe --> it should detections because this can be dependent on the detection efficiency
4-figure 11: please specify in the figure what are the dashed and continous lines
5-page 24: O(1) times larger --> does it mean it is of the same level? please clarify
6-page 25: "side effect" radiological backgrounds --> usually referred to as muon-induced background or cosmogenic backgrounds
7-page 27: of the atomic electrons --> off the atomic electrons
8-page 29: nuclear recoil in an ultra-cold germanium crystal --> please cite some example experiment
9-page 30: superheated fluid-> an example here would better clarify
10-page 33: our airborne radioisotopes --> out airborne radioisotopes
11-page 33: to passively reduce --> or to moderate neutrons
12-page 33: radiognic -->radiogenic
13-page 33: it is worth to mention also Gd-loaded water as active shield for neutrons (used by XENONnT)
14-page 39:The XENON collaboration columns are used to purify the liquid Xenon from both Krypton and Radon with an "online" distillation system
15-page 40: heavy targets like germanium -> please cite some experiment
16-page 40: please specify how the atmospheric neutrinos are expected to interact
17-page 42: directional detection: it would be interesting to discuss the level of sensitivity the experiments under construction could reach
18-page 43: long exposures : this translates in the need of very large volumes
19-figure 25: the figures are too small to read. Please magnify them.
20-figure 25: it would be interesting to mention in the text what are the main theoretical models that are represented in the figures
21-page 47: avrage --> average
22-page 48: Please cite also the SABRE project and the interest to have a dual detectors in both emispheres
page 48: LUX/LZ: it is now just LZ
page 48: known drift of ions: this is done by using the time drift information from the difference between S1 and S2
23-page 48: further accelerated: thanks to a stronger electric field
24-page 52: is s strong --> is a strong
25-page 55:are sensitivity --> are sensitive
26-page 56: evetns -->events
27-page 58: there is a typo with a link adress (overleaf) to remove
28-page 58: E basis --> ER basis
29-page 58: Edelweiss: please specify the LSM location
30-page 58: please specify why the resolution and backgrounds from "heat-only" events are limiting factors
31-page 61: TEX --> TES
32-page 62: Figure 42 is not correctly referenced
33-page 63: figure 42: specify the legend for the various plots in the figure
34-page 62: there is no reference to figure 43 in the text
35-page 65: Conclusions --> from the conclusion it is not clear what is the maximal sensitivity for G2 experiments. Please clarify that the neutrino fog will play a role for G3 experiments.
35-page 66: LUX --> No, LZ

---

## Round 2 · List of Changes

Changes in response to Reviewer 1:
1-general: the Sun when referred to the name of the star in our Solar System should be written in a capital letter
2-page 10 the generates --> that generates
3-page 22: about-->around
3-page 23: interactions to observe --> it should detections because this can be dependent on the detection efficiency
sentence has been re-worded
4-figure 11: specified in the figure what are the dashed and continous lines represent
5-page 24: O(1) times larger --> does it mean it is of the same level? please clarify
sentence re-worded
6-page 25: changed "side effect" radiological backgrounds to as muon-induced background or cosmogenic backgrounds
7-page 27: of the atomic electrons --> off the atomic electrons
10-page 33: our airborne radioisotopes --> out airborne radioisotopes
11-page 33: to passively reduce --> or to moderate neutrons
12-page 33: radiognic -->radiogenic
14-page 39: added radon to the list of contaminants that are reduced by distillation techniques in liquid nobles
16-page 40:discussion of cosmic ray interatctions
19-figure 25: increased figure size
20-figure 25: added list of models in figure 25 to the main text.
21-page 47: avrage --> average
22-page 48: Mentioned the SABRE project and the interest to have a dual detectors in both emispheres
page 48: changed LUX/LZ: to LZ
page 48: clarified time drift information between S1 and S2
23-page 48: further accelerated through a stronger electric field
24-page 52: is s strong --> is a strong
25-page 55:are sensitivity --> are sensitive
26-page 56: evetns -->events
27-page 58: fixed typo with a link address (removed the link address)
28-page 58: E basis --> ER basis
29-page 58: specify the location of LSM
30-page 58: discuss briefly why the resolution and backgrounds from "heat-only" events are limiting factors
31-page 61: TEX --> TES
32-page 62: fixed Figure 42 refernece
33-page 63: specified the legend in figure 42
34-page 62: removed figure 43
Changes in response to reviewer 2:
Page 9 fixed typo "detector" -> "detect"
Changes in response to reviewer 3:
1 - added more references on page 4 to discussion of dark matter distribution.
2 - revised text to referring to WIMP
3 - page 5: Note in paragraph following equation 7 that inelastic scattering is possible and added reference.
4 - pg 15: Paragraph following equations 87-88 summarizes the situations under which those equations are valid.
1-general: the Sun when referred to the name of the star in our Solar System should be written in a capital letter
2-page 10 the generates --> that generates
3-page 22: about-->around
3-page 23: interactions to observe --> it should detections because this can be dependent on the detection efficiency
sentence has been re-worded
4-figure 11: specified in the figure what are the dashed and continous lines represent
5-page 24: O(1) times larger --> does it mean it is of the same level? please clarify
sentence re-worded
6-page 25: changed "side effect" radiological backgrounds to as muon-induced background or cosmogenic backgrounds
7-page 27: of the atomic electrons --> off the atomic electrons
10-page 33: our airborne radioisotopes --> out airborne radioisotopes
11-page 33: to passively reduce --> or to moderate neutrons
12-page 33: radiognic -->radiogenic
14-page 39: added radon to the list of contaminants that are reduced by distillation techniques in liquid nobles
16-page 40:discussion of cosmic ray interatctions
19-figure 25: increased figure size
20-figure 25: added list of models in figure 25 to the main text.
21-page 47: avrage --> average
22-page 48: Mentioned the SABRE project and the interest to have a dual detectors in both emispheres
page 48: changed LUX/LZ: to LZ
page 48: clarified time drift information between S1 and S2
23-page 48: further accelerated through a stronger electric field
24-page 52: is s strong --> is a strong
25-page 55:are sensitivity --> are sensitive
26-page 56: evetns -->events
27-page 58: fixed typo with a link address (removed the link address)
28-page 58: E basis --> ER basis
29-page 58: specify the location of LSM
30-page 58: discuss briefly why the resolution and backgrounds from "heat-only" events are limiting factors
31-page 61: TEX --> TES
32-page 62: fixed Figure 42 refernece
33-page 63: specified the legend in figure 42
34-page 62: removed figure 43
Changes in response to reviewer 2:
Page 9 fixed typo "detector" -> "detect"
Changes in response to reviewer 3:
1 - added more references on page 4 to discussion of dark matter distribution.
2 - revised text to referring to WIMP
3 - page 5: Note in paragraph following equation 7 that inelastic scattering is possible and added reference.
4 - pg 15: Paragraph following equations 87-88 summarizes the situations under which those equations are valid.

---

## Editorial Decision

published